# A stratospheric precursor of East Asian summer droughts and floods

Ruhua Zhang[1], Wen Zhou [ORCID][1,2] ✉, Wenshou Tian [ORCID][3], Yue Zhang[1], Junxia Zhang[4] & Jiali Luo[3]

East Asian floods and droughts in summer show a typical dipole pattern with a north-south oscillation centered near 30°N, called the southern drought–northern flood (SDNF) pattern, which has caused significant economic losses and casualties in the past three decades. However, effective explanations and predictions are still challenging, making suitable disaster prevention more difficult. Here, we find that a key predictor of this dipole pattern is the Quasi-Biennial Oscillation (QBO, tropical winds above 10 km). The QBO can modulate precipitation in East Asia, contributing the largest explained variation of this dipole pattern. A QBO-included statistical model can effectively predict summer floods and droughts at least three months in advance and explain at least 75.8% of precipitation variation. More than 30% of the SDNF pattern is attributed to the QBO in July-August 2020 and 2021. This result suggests a good prospect for using the tropical mid- to upper atmosphere in seasonal forecasts for summer.

Extreme precipitation, drought and flood disasters often occur in summer in East Asia, one of the world's economic centers and most densely populated areas, resulting in significant economic damage and casualties[1–3]. For instance, the 1998 Yangtze flood in China caused approximately 3000 deaths and about 30 billion dollars in damage[3,4]. Extensive flooding during June–July 2020 caused at least 141 deaths in China and 46 deaths in South Korea[5–8]. The recent extreme drought in 2022 led China's largest freshwater lake and longest river to dry up, and about 450 million people had to deal with depleted wells as well as brush fires[9]. Thus, a wide consensus has been reached, by both scientists and the public, that accurate seasonal forecasts of East Asian summer precipitation (EASP) could give us more time to prepare to manage water resources and reduce drought and flood disasters[10–12].

EASP is regulated by the East Asian summer monsoon (EASM) system, one of the most active monsoon systems in the world[13–15]. When the EASM advances, the rain belt gradually moves northward through several stages[13]. The rainy season usually arrives in East Asia in June-August; it is called Meiyu in China, Changma in Korea, and Baiu in Japan[13,14]. The distribution of EASP shows strong inter-annual variability

and largely determines whether droughts and floods occur. The leading empirical orthogonal function (EOF1) mode of EASP has three anomalous centers in the meridional direction before the mid-1990s, but it becomes a meridional dipole mode centered on the Yangtze River after the mid-1990s. This meridional dipole is called the southern flood–northern drought (SFND) or southern drought–northern flood (SDNF) pattern[16–18].

Many studies have explored the causes of summer precipitation and the prediction of droughts and floods over East Asia from the perspective of air-sea interactions, land-air processes, and tropospheric internal dynamics[12,17,19]. The research on air-sea interactions has a wide spatial span and involves major marine signals in four oceans, such as the well-known El Niño–Southern Oscillation (ENSO)[20–22], Pacific Decadal Oscillation (PDO)[20,23–25], Indian Ocean Dipole (IOD)[5,10,26,27], Atlantic sea surface temperature[19,25,28,29], and Arctic sea ice[19]. Land-air processes are centered on the Tibetan Plateau, the highest plateau in the world[27,29]. Atmospheric internal dynamics involve various systems in the lower atmosphere, such as the Western North Pacific Subtropical High (WNPSH)[30,31], South Asian High[32,33], and

[1]Key Laboratory of Polar Atmosphere-ocean-ice System for Weather and Climate, Ministry of Education, Department of Atmospheric and Oceanic Sciences and Institute of Atmospheric Sciences, Fudan University, Shanghai, China. [2]Key Laboratory for Polar Science of the MNR, Polar Research Institute of China, Shanghai, China. [3]Key Laboratory for Semi-Arid Climate Change of the Ministry of Education, College of Atmospheric Sciences, Lanzhou University, Lanzhou, China. [4]Lanzhou Central Meteorological Observatory of Gansu Province, Lanzhou, China. ✉e-mail: wen_zhou@fudan.edu.cn

North Atlantic Oscillation[28,34]. However, significant challenges remain in understanding and predicting EASP, with both statistical and dynamic models[12,17]. This means that some important factors or mechanisms have not yet been recognized, such as the Quasi-Biennial Oscillation (QBO, tropical middle to upper atmospheric zonal wind above 10 km), which is usually neglected in most statistical and dynamic models[12,35].

Recent studies have found that weather and climate variations over East Asia in winter are controlled by the QBO to a certain extent[35,36]. For instance, the easterly phase of QBO (EQBO) tends to bring a warm winter to East Asia, which is achieved mainly by regulating the subtropical jet stream and the East Asian winter monsoon (EAWM)[35,36]. The QBO-induced temperature changes can modulate static stability near the tropical tropopause and further the Madden-Julian Oscillation (MJO)[37]. The difference in precipitation anomalies between the EQBO and westerly phases of QBO is ~70% in MJO phases 6–8 from southern China to Japan. The linkage between the QBO and precipitation in southeast China is controlled by the Holton-Tan effect (QBO's influence on the stratospheric polar vortex) in winter; a significant QBO-precipitation relationship occurs mainly when the Holton-Tan effect is weak[38]. Compared to winter, the influences of the QBO in summer are not receiving enough research attention. Some studies have found that the QBO signals exist in precipitation in some regions of East Asia, such as the Huaihe River valley[39,40] and August precipitation in northern China[41]. However, we still do not know

whether the QBO is important in the leading mode of EASP and its mechanism. Therefore, the objective of this study is to investigate the response of EASP to the QBO and the relevant mechanisms after 1995. We also diagnose the relative contribution of the QBO to the dipole mode and use the QBO to predict EASP in 2020 and 2021. Our results demonstrate the key role of the QBO in precipitation change over East Asia, providing a good insight into understanding and predicting EASP.

## Results

### East Asian summer precipitation

Affected by the complex EASM system, EASP is controlled by various major impacting factors during different states. In this study, we explain mainly precipitation and its meridional dipole mode in July and August, when the rain belt usually arrives in the Yangtze River Basin, North China, and the Korean Peninsula[13,14]. Figure 1 shows East Asian precipitation and the Standardized Precipitation Index (SPI) in July–August. High precipitation is observed over Eastern China and the Korean Peninsula (Fig. 1a, Supplementary Fig. 1a, b), areas with frequent drought and flood disasters[1,2]. EOF1 of precipitation and the SPI shows a southern flood–northern drought pattern centered near 30°N (Fig. 1b, c, Supplementary Fig. 1c, d), as reported in previous studies[17,18]. In order to conveniently represent the north-south oscillations of East Asian precipitation, we establish a dipole mode index (DMI) and a corresponding dipole mode index for SPI (DMI-SPI, deep blue lines in Fig. 1d) using the differences between the weight-averaged anomalies

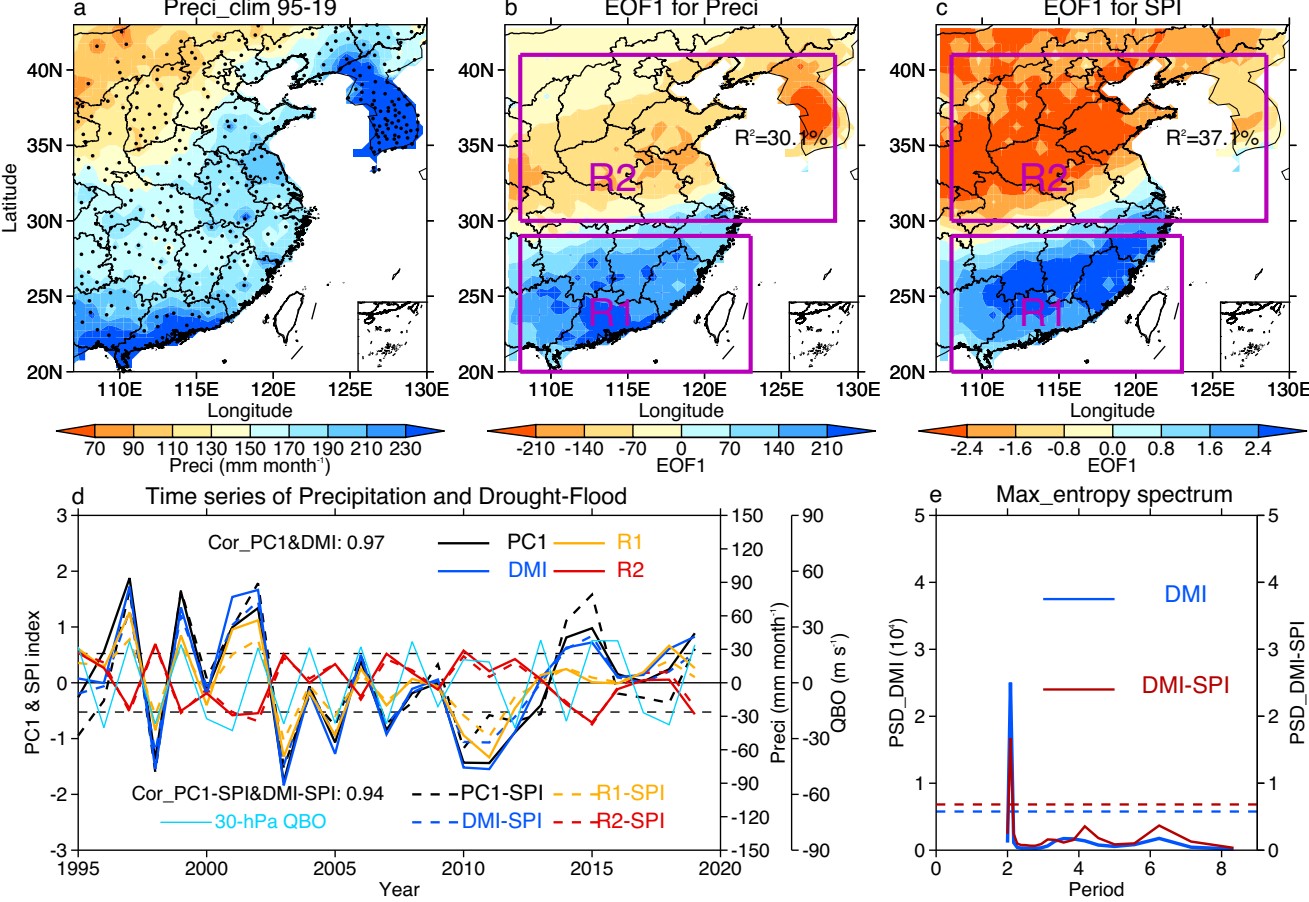

**Fig. 1 | East Asian precipitation (107–130°E, 20–43°N) in July and August over the period 1995–2019. a** Spatial distribution of climatological precipitation (unit: mm month⁻¹). **b** The leading empirical orthogonal function (EOF1) pattern for precipitation. **c** As in (**b**) but for Standardized Precipitation Index (SPI). **d** Weight-averaged anomalies over R1 (108-123°E, 20–29°N, orange) and R2 (108-128°E, 30-41°N, red), the dipole mode index (DMI = R1–R2, deep blue), standardized EOF1

time series (PC1, black), and the QBO index anomalies (light blue). **e** Maximum entropy spectrum for the DMI and DMI of SPI (DMI-SPI). Black dots in (**a**) indicate station locations. Solid and dashed lines in **d** show the precipitation and SPI, respectively. Dashed straight lines in (**e**) are the white noise test at the 95% confidence level.

in the southern region (R1, 108-123°E, 20-29°N) and northern region (R2, 108-128°E, 30-41°N). The DMI/DMI-SPI can rebuild the variations in the EOF1 time series of precipitation/SPI (PC1/PC1-SPI, black lines in Fig. 1d) with a high correlation coefficient of 0.97/0.94. Note that the DMI and DMI-SPI have a pronounced spectral peak of nearly 2 years (Fig. 1e), suggesting that variations in the dipole mode are controlled by impacting factors with a quasi-biennial cycle, such as sea surface temperature (SST) signals and the QBO (Fig. 1d, light blue line).

### Influence of sea surface temperature

Figure 2 shows the circulation differences (SFND minus SDNF) and correlation coefficients between the DMI and detrended SST anomalies around the world. The WNPSH is a crucial system with a profound impact on the East Asian summer climate[42–44]. Compared to SDNF, SFND is usually associated with a weaker WNPSH (plus signs, Fig. 2a) and an eastward WNPSH western ridge point (triangles, Fig. 2a). Cyclonic anomalies transport more warm/cold air south/north of 30°N (Supplementary Fig. 2a–c), leading to anomalous ascending/sinking motion there (Fig. 2b). Meanwhile, water vapor anomalously converges south of 30°N and diverges north of 30°N (Fig. 2c, Supplementary Fig. 2d–f). According to the water vapor budget equation, these processes directly determine precipitation anomalies[45–48].

SST is a traditional research field for the WNPSH and EASP, covering most areas of the four oceans and the timescales involved in the

contemporaneous, cross-seasonal, inter-annual, and decadal relationships[5,19,21,22,26–28]. Based on previous studies, we briefly focus on several SST factors, with two indicators for seasonal forecasts. One is the direct contemporaneous relationship between EASP and SST anomalies. The other is the ability to trace the EASP-SST relationship back to SST changes a few months before. This considers the fact that SST anomalies usually exist longer than atmospheric processes, and SST signals from a few months before can continue into July and August[25]. Although some impacting factors, such as IOD and ENSO, have cross-seasonal influences[21,22], they are not included in this study due to their indirect effects, complexity, and uncertainty.

Significant contemporaneous linkages can be observed between EASP and some ocean areas (Fig. 2a and Supplementary Fig. 3), including the north tropical Atlantic (NTA), subpolar North Atlantic (SNA), south Indian Ocean (SIO), and tropical Pacific (ENSO-like signals), which have been widely demonstrated in previous studies[25]. There are several possible ways in which Atlantic SST signals can influence EASP. For instance, the North Atlantic SST may exert a westward influence on the Pacific and the East Asian climate[25]. Summer NTA SST can impact the WNPSH via a zonally anomalous circulation over the North Pacific–North Atlantic (Supplementary Fig. 4a, b)[49]. North Atlantic SST can also impact the downstream Eurasian climate via Rossby wave trains traveling eastward from the North Atlantic to East Asia (Supplementary Fig. 4c, d)[25,28]. Indian Ocean SST plays an

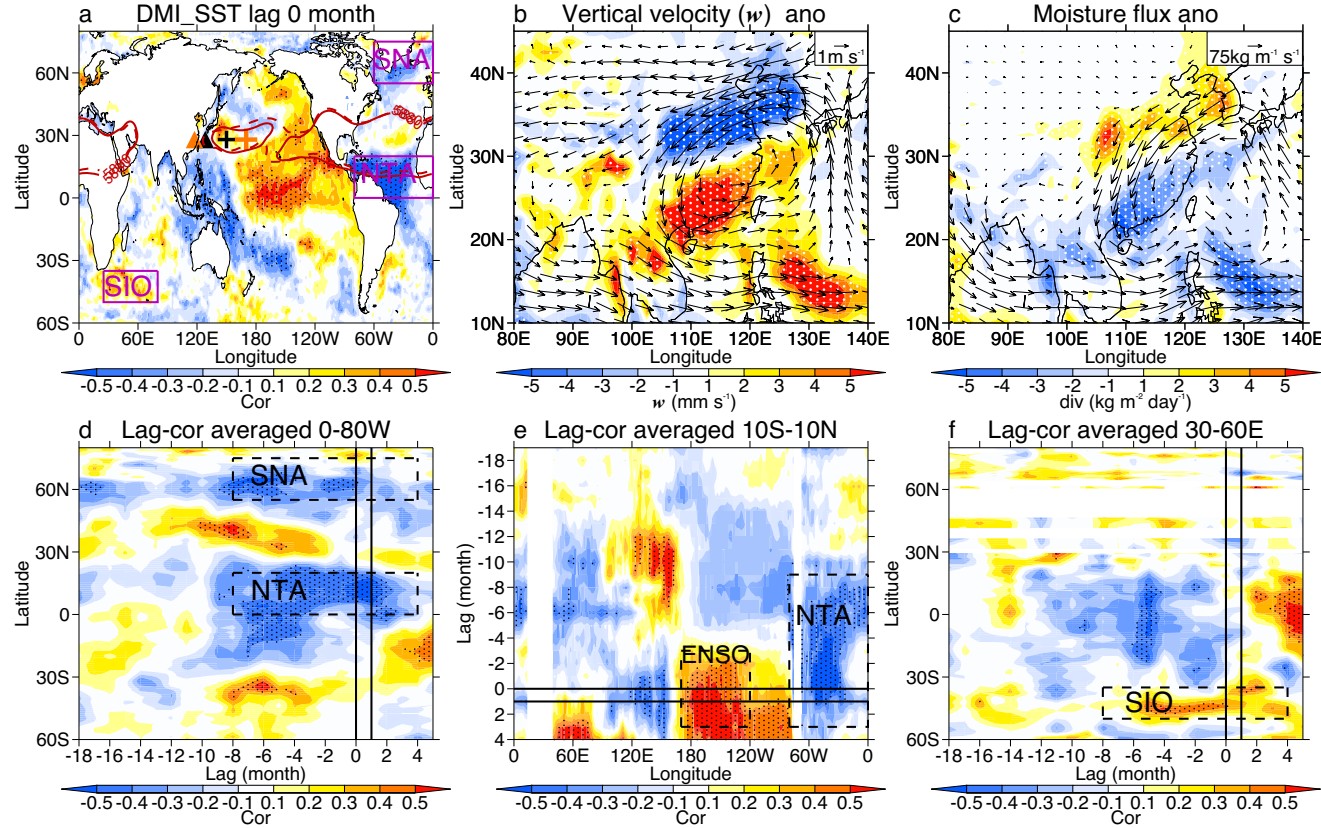

**Fig. 2 | Linkages between the dipole mode index (DMI) and detrended sea surface temperature (SST)/circulation anomalies over the period 1995–2019.** **a** Simultaneous correlation coefficient in the global ocean. **b** Composite differences (southern flood–northern drought year/SFND minus southern drought–northern flood year/SDNF) of horizonal circulation (vectors) and vertical velocity (colors) anomalies at 500 hPa. **c** As in (**b**) but for vertically integrated moisture flux (vectors) and its divergence (colors) anomalies. **d** Lag correlation over the Atlantic region (averaged over 0–80°W). **e** Lag correlation over the tropical region (averaged over 10°S-10°N). **f** Lag correlation over the Indian Ocean (averaged over 30-60°E). Solid and dashed contours in (**a**) are the 5880-gpm

geopotential height during SFND and SDNF years, respectively. The locations of the western ridge points of the Western North Pacific Subtropical High (WNPSH) and strength of the WNPSH are indicated by triangles and plus signs in (**a**), respectively. Orange/black triangles and plus signs represent the SDNF/SFND years, and a larger plus sign means a stronger WNPSH. The NTA, SNA, and SIO indices are the weight-averaged SST anomalies over 0–80°W and 0–20°N, 0–60°W and 55–75°N, and 25–80°E and 35–50°S, respectively. Dashed boxes in (**d–f**) show the locations of NTA, SNA, and SIO. Straight lines in (**d–f**) are July and August, and dots indicate values at the 95% confidence level according to Student's t-test.

important role in the development of the Asian summer monsoon system, which has both contemporaneous and cross-seasonal influences[25,27]. South Indian Ocean SST anomalies drive a strong Mascarene High, which results in a cross-equatorial flow, a weak WNPSH, and anomalous vertical motion over East Asia (Supplementary Fig. 5a, b)[50]. Indian Ocean SST can also lead to a cross-seasonal effect on EASP by modulating Tibetan Plateau snow and equatorial Pacific SST[21,22,27]. Note that NTA, SNA, and SIO SST anomalies can be traced back to at least 6 months before (Fig. 2d–f, Supplementary Fig. 3), suggesting that these signals may be used to improve seasonal forecasting skills. The ENSO-like SST has significant impacts on EASP[21,22]; however, its contemporaneous influence can be traced back to only 2 months before the DMI (Fig. 2e, Supplementary Fig. 3). Thus, it is not considered here, although there are also some cross-seasonal effects.

## Role of the Quasi-Biennial oscillation

The strong 2-year spectral peak suggests a possible modulation of the QBO on EASP, which has not previously been recognized. Figure 3 shows the QBO-related and DMI-related circulation anomalies. In the tropical lower stratosphere, DMI-related westerly anomalies are observed, which arch downward to the subtropical troposphere in the Northern Hemisphere (Fig. 3a). These downward-arching westerly anomalies combined with mid-latitude easterly anomalies establish a dipole mode centered near 30°N in the troposphere, corresponding to the EOF1 pattern in Fig. 1b, c. This DMI-related zonal wind pattern is very similar to the QBO-induced zonal wind anomalies (colors in Fig. 3a, b), further suggesting that a QBO signal exists in the DMI changes. How, then, could the QBO modulate precipitation over East Asia? Previous studies have pointed to three possible ways in which the QBO can impact the surface: by influencing the stratospheric polar vortex, by impacting tropical convective activity, and by modulating the subtropical jet and wave activity[35,36,51,52]. The polar way is invalid because there is no stratospheric polar vortex in the summer hemisphere, and the responses of zonal wind are weak in the tropical troposphere (Fig. 3b, Supplementary Fig. 6a); thus, we investigate mainly the subtropical way here.

The subtropical way involves two critical processes: the QBO-induced secondary circulation (QBO's direct effect) and the corresponding tropospheric wave-flow feedback (enhancing tropospheric QBO signals), according to a previous study[52]. The tropical Kelvin, intertia-gravity, Rossby-gravity and gravity waves force the QBO signals to propagate downward from the upper to the lower stratosphere. To maintain momentum and thermal wind balance, the QBO induces secondary circulations from the tropical upper troposphere to the upper stratosphere, with a negative stream function anomaly near 30 hPa in the Northern Hemisphere and two positive stream function anomalies on its upper and lower sides (Fig. 3c). These positive and negative stream function anomalous centers propagate downward from the upper stratosphere into the upper troposphere, following the downward propagation of QBO (Fig. 3d). In the Northern Hemisphere, a positive/negative stream function means an anticyclonic (clockwise)/cyclonic (anticlockwise) circulation. According to the relationship between the zonal wind and stream function, westerly anomalies are located on the north side of positive stream function anomalies. Thus, with positive stream function anomalies appearing in the tropical upper troposphere (Fig. 3c, d), westerly anomalies occur in the subtropical upper troposphere over East Asia (colors in Fig. 3b)[52]. According to the equation of relative vorticity, meridional gradient of zonal wind will increase (reduce) the relative vorticity on the north (south) side of these subtropical westerly anomalies (Fig. 3e). The abnormal relative vorticity could disturb the anomalous meridional vortex stretching and meridional advection of absolute vorticity term of Rossby wave source (RWS, see Fig. 3f and method), which further cause a positive RWS anomaly in the subtropical region and poleward Rossby wave flux anomalies (Fig. 3g). The wave-induced momentum

fluxes anomalously diverge in the subtropics and converge in the mid-latitudes (Supplementary Fig. 6b), amplifying the subtropical westerly anomalies and forcing mid-latitude easterly anomalies (colors, Fig. 3h). Based on the geostrophic adjustment theory, zonal wind anomalies can result in a cyclonic anomaly over East Asia (contours, Fig. 3h). Such RWS and wave flux anomalies occur mainly in the upper troposphere (above 300 hPa, Supplementary Fig. 6c–h), suggesting that the wave-flow feedback is an upper tropospheric process induced by the QBO.

The QBO-induced cyclonic anomaly extends from the stratosphere to the surface, accompanied by a cold center in the troposphere over East Asia (Fig. 4a, b). Relevant horizonal wind and temperature anomalies lead to changes in temperature advection (Supplementary Fig. 7a–c). These temperature advection anomalies further drive a south-north oscillation pattern in the vertical motion, with anomalous ascending/sinking motion south/north of 30°N (Fig. 4c, d). Moisture flux divergence also shows a similar dipole mode with anomalous convergence/divergence south/north of 30°N (Fig. 4e, Supplementary Fig. 7d–f). Finally, the moisture flux and vertical motion together drive anomalous precipitation over East Asia (Fig. 4f). To verify the linkages between EASP and the stratosphere, 15 experiments nudged to the historical stratosphere are designed using different initial conditions (see methods and Supplementary Table 1). Each experiment runs for 6 months every year over the period of 1995–2019. Because some researchers believe that the QBO signals in EASP may be related to traditional factors such as ENSO (Supplementary Table 2), external forcings including SST are set at the climatological values to avoid such noise. Experimental results produce a QBO-induced zonal wind pattern similar to the observations (Figs. 4g, 3h). A meridional dipole mode is also observed in the experimental precipitation over East Asia (Fig. 4h), indicating the importance of QBO in EASP. Usually, the QBO-induced zonal wind shows an evident downward propagation from the upper stratosphere to the troposphere (Fig. 4i, Supplementary Fig. 8a); thus, the mid- to upper stratospheric QBO could also precede DMI changes in July–August (Supplementary Fig. 8b), indicating a possibility that the QBO could improve seasonal forecasts of EASP.

## Seasonal forecasts of EASP using the QBO

The analyses above have shown the importance of QBO and SST in the leading mode of EASP (Schematic diagram in Fig. 5a), but we still need to assess the relative contributions of these factors to seasonal forecasts. We therefore establish a seasonal forecasting equation and diagnose the contributions of the above factors. The 10-hPa QBO in March has a strong connection with the 30-hPa QBO and DMI in July-August (Supplementary Fig. 8b, c), which is chosen as a predictor. The propagation speed of QBO is not a constant from 10 hPa to the tropopause; thus, the 70-hPa QBO with the portion linked to the 10-hPa QBO removed is also used to adjust the propagation speed of QBO. The SST anomalies over NTA, SNA, and SIO are also selected in March. A multiple linear regression equation is established using the datasets over 1995-2019 to predict EASP in 2020 and 2021 (see methods). The regression equation has a high correlation coefficient of 0.87 with the DMI time series and explains 75.8% of the variance of the DMI (Fig. 5b). Particularly, the QBO accounts for more than 40% of the explained variance, which has the largest contribution of all factors. In addition, the QBO contributes more than 30% of the DMI in 2020 and 2021 (Fig. 5c). Forecasting results of the SPI are similar to the precipitation anomalies (Supplementary Fig. 9), suggesting the ability of our model to predict floods and droughts.

Although these predictors are chosen based on the DMI time series, they are also useful in seasonal forecasts of precipitation in most spatial grids (Fig. 5d). The model can predict the major spatial distribution of EASP in 2020 and 2021, except for the Korean Peninsula and the northwest region of East Asia, which have low explained variance in our seasonal forecasting model (Fig. 5e, f, Supplementary

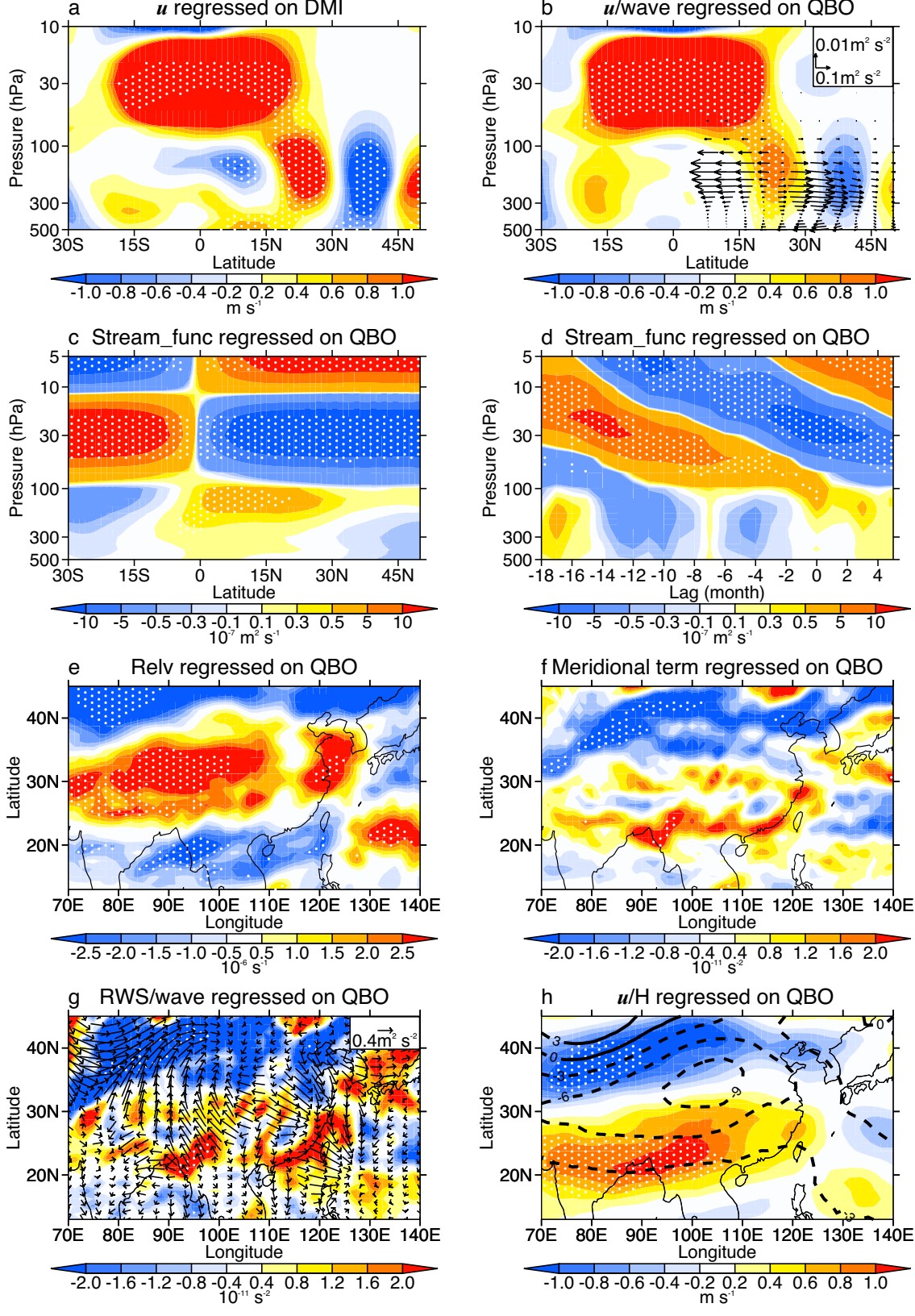

Fig. 10a–c). The precipitation anomalies in 2022 are not a dipole mode near 30°N like EOF1 (Supplementary Fig. 10d). Our model can also predict some of the droughts and floods in 2022, and the QBO contributes positively to droughts between 27°N and 35°N (Supplementary Fig. 10e, f). To summarize, the QBO is a key factor in EASP and its leading mode, which has previously been overlooked. A better

understanding of QBO could give us a good insight into predicting EASP.

## Discussion

Droughts and floods are two of the most severe natural disasters in summer in East Asia; however, their seasonal forecasts are still

**Fig. 3 | Detrended circulation anomalies regressed on the dipole mode index (DMI) and 30-hPa Quasi-Biennial Oscillation (QBO) index over the period 1995-2019. a** East Asian zonal wind anomalies (colors, averaged over 70–120°E) regressed on the standardized DMI. **b** East Asian zonal wind (colors) and Plumb wave flux (vectors) anomalies averaged over 70–120°E regressed on the standardized QBO index. **c** Zonal mean stream function anomalies (colors) regressed on the standardized QBO index. **d** Tropical zonal mean stream function anomalies (colors, averaged over 0–15°N) lag-regressed on the standardized QBO index. **e** The 200-hPa relative vorticity anomalies regressed on the standardized QBO index.

**f** As in (**e**) but for the meridional term of 200-hPa Rossby wave source (RWS). **g** The 200-hPa RWS (colors) and Plumb wave flux (vectors) anomalies regressed on the standardized QBO index. **h** The 200-hPa zonal wind (colors) and geopotential height (contours) anomalies regressed on the standardized QBO index. The units of geopotential height are gpm in (**h**). NTA, SNA, and SIO indices are the weight-averaged SST anomalies over 0–80°W and 0–20°N, 0–60°W and 55–75°N, and 25–80°E and 35–50°S, respectively. Linear effects from NTA, SNA, and SIO have been removed in the anomalies in (**b**–**h**). Dots indicate values at the 95% confidence level according to Student's *t*-test.

challenging due to unclear mechanisms. Traditionally, the dipole mode pattern of EASP has usually been attributed to both the contemporaneous and cross-seasonal effects of SST and lower atmospheric dynamics[12,17,19,21,22], and few studies have recognized the importance of the middle and upper atmosphere above 10 km in the variation of EASP. Here, we emphasize the role of the tropical stratospheric QBO in droughts and floods over East Asia. Both observational and modeling analyses demonstrate a strong modulation of QBO on the leading mode of EASP.

A statistical model is established, based on the joint effects of QBO and traditional SST. Compared with traditional impacting factors, the QBO has the largest explained variance in the DMI and contributes more than 30% of the SDNF mode in 2020 and 2021. Note that the QBO has a roughly 2-year period, however, it causes a consistent spatial distribution of the SDNF mode in 2020 and 2021. This is because the easterly wind continues for a short while in 2020 and a QBO disruption occurs in 2020–2021 (Supplementary Fig. 8a). Thus, the easterly wind appears for two consecutive years. Our statistical model is based on factors involving both the contemporaneous and lagged influences on EASP, which can explain only part of the DMI in 2020 and 2021 (Fig. 5b). In fact, other factors are also important, such as cross-seasonal factors and synoptic-scale processes in individual years. For instance, the extensive flooding in 2020 was partly a result of the IOD, an important cross-seasonal factor[5,21]. Typhoons Cempaka and In-Fa remotely controlled the extreme flooding in Henan, China, in 2021[53].

In contrast to the currently prevailing stratospheric research concentrated on the polar vortex in the winter hemisphere[54], this study considers the middle and upper atmosphere in seasonal forecasts in the summer hemisphere, when the stratospheric polar vortex is absent. Here, we have displayed the importance of the QBO period-phase in EASP forecasts. However, the roles of other QBO characteristics such as vertical–horizontal structure and amplitude are still unclear, and these may also be important in the weather and climate over East Asia[51,55]. These factors need to be better incorporated into seasonal forecasting in the future.

## Methods
### Datasets
This study uses daily precipitation datasets from national standard meteorological stations in China and the Korean Peninsula over the period of 1990−2021. The precipitation in China has been quality-controlled and checked by the National Meteorological Information Center (NMIC). We estimate the missing data and check the homogeneity at each station using Zhang's methods[56]. The missing data are estimated using the inverse distance method. Four statistical methods are used to check the homogeneity of these datasets, including the standard normal homogeneity test, moving *t*-test, penalized maximal *t*-test and Buishand range test. A time series is inhomogeneous when all the methods reject the null hypothesis. Finally, we retain 523 stations that pass the homogeneity test at the 95% confidence level. The daily precipitation in the Korean Peninsula includes 60 stations in South Korea and 27 stations in North Korea. These station data are

interpolated onto 0.5° × 0.5° latitude-longitude grids using the inverse distance method. The western ridge points and strength of the WNPSH are from the National Climate Centre in China[57,58].

The Japanese 55-year (JRA-55, 1.25° latitude × 1.25° longitude) reanalysis datasets are also analyzed over the period 1995−2019[59], including monthly mean temperature (*T*), zonal wind (*u*), meridional wind (*v*), stream function, and omega etc. SST is obtained from the Met Office Hadley Centre Sea Ice and Sea Surface Temperature Dataset Version 1 (HadISST1, 1.0° latitude × 1.0° longitude)[60]. The observational QBO indices are from the Singapore Sonde monthly mean winds.

### Analytical methods
Anomalies for all variables are computed as deviations from the seasonal climatological values (averaged over 1995−2019), and linear trends have been removed in this study. We use the least squares method to calculate the linear regression and a two-sided Student's *t*-test to diagnose statistical significance. Effective degrees of freedom in the correlation analysis are computed based on one-lag autocorrelation[61]. A power spectrum is estimated by the maximum entropy method[62]. The SPI is adopted to diagnose meteorological droughts and floods[63]. The propagation of waves in three-dimensional space is diagnosed using Plumb wave flux (equation 5.7 in Plumb's work)[64].

$$F_s = p \cos\varphi \begin{pmatrix} \frac{1}{2a^2\cos^2\varphi}\left[\left(\frac{\partial\psi'}{\partial\lambda}\right)^2 - \psi'\frac{\partial^2\psi'}{\partial\lambda^2}\right] \\ \frac{1}{2a^2\cos\varphi}\left(\frac{\partial\psi'}{\partial\lambda}\frac{\partial\psi'}{\partial\varphi} - \psi'\frac{\partial^2\psi'}{\partial\lambda\partial\varphi}\right) \\ \frac{2\Omega^2\sin^2\varphi}{N^2 a\cos\varphi}\left(\frac{\partial\psi'}{\partial\lambda}\frac{\partial\psi'}{\partial z} - \psi'\frac{\partial^2\psi'}{\partial\lambda\partial z}\right) \end{pmatrix} \quad (1)$$

Where $F_s$, $\psi$, $a$, $\varphi$, $\lambda$, $z$, $N$, and $\Omega$ are the wave flux, stream function, Earth radius, latitude, longitude, altitude, buoyancy frequency, and Earth's rotation. $p$ = pressure/1000 hPa. The primes indicate the deviations from the zonal mean.

### Definition of Index
The DMI and DMI-SPI are defined as the differences between weight-averaged anomalies in the southern region (R1, 108-123°E, 20–29°N) and northern region (R2, 108-128°E, 30–41°N). SFND (SDNF) events occur when the standardized DMI is greater than 0.5 (less than -0.5). The QBO index is defined as tropical zonal wind in observations. The NTA, SNA, and SIO indices are the area-averaged SST anomalies over 0-80°W & 0-20°N, 0-60°W & 55-75°N, and 25-80°E & 35-50°S, respectively. The Niño-3.4 index is computed as area-averaged SST anomalies (5°S-5°N, 170°-120°W). The southern annular mode index is defined as the difference in surface pressure between 40°S and 65°S.

### Temperature advection
Temperature advection (*VT*, hereafter) is calculated as follows:

$$VT = -V \cdot \nabla T = -u \cdot \frac{\partial T}{\partial x} - v \cdot \frac{\partial T}{\partial y} \quad (2)$$

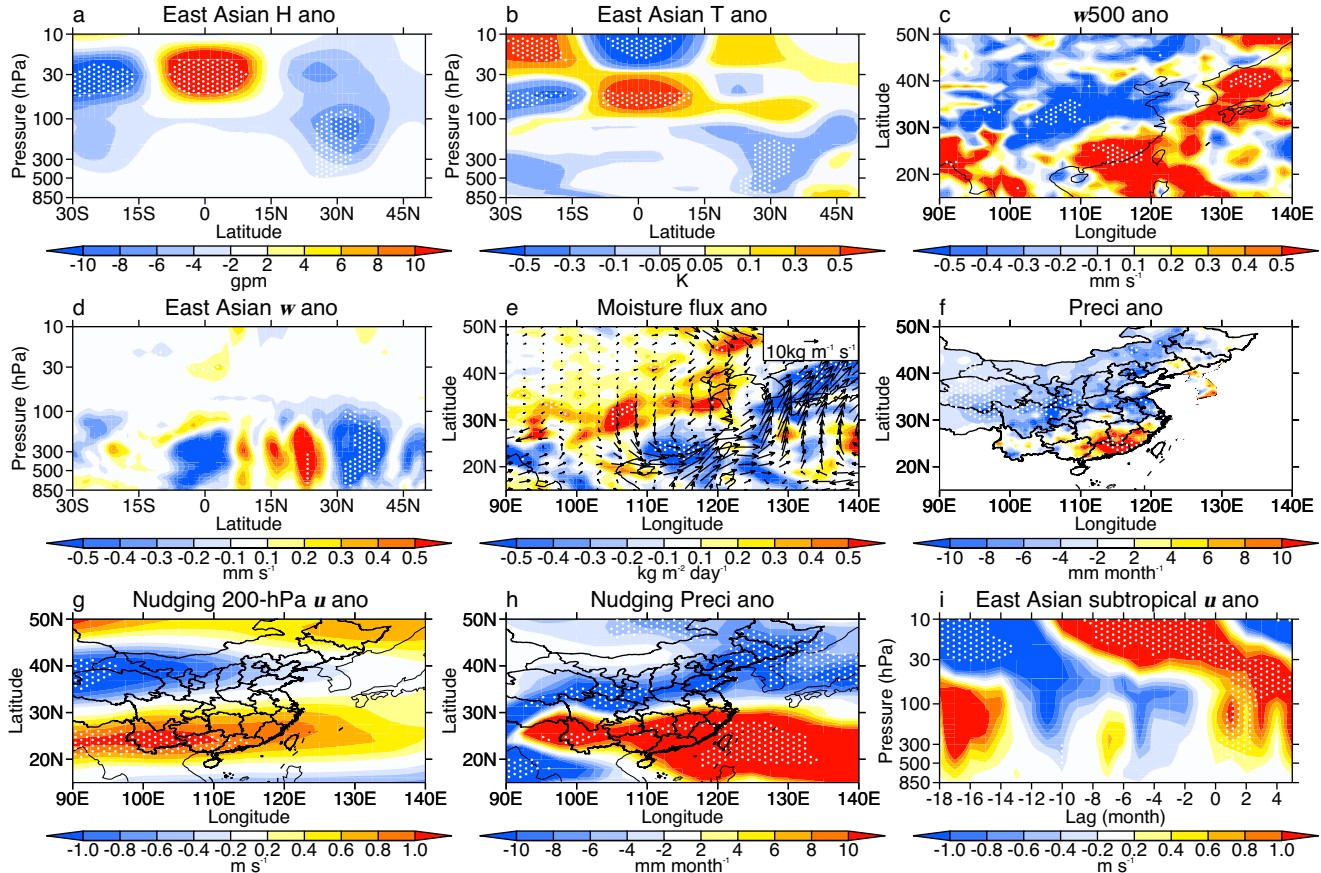

**Fig. 4 | East Asian detrended circulation anomalies regressed on the 30-hPa standardized Quasi-Biennial Oscillation (QBO) index over the period 1995–2019. a** East Asian geopotential height anomalies (colors, averaged over 70–120°E). **b** As in (**a**) but for the temperature anomalies (colors). **c** The 500-hPa vertical velocity anomalies (colors). **d** Vertical velocity anomalies averaged over 105-120°E (colors). **e** Vertically integrated moisture flux (vectors) and its divergence (colors) anomalies. **f** Observational precipitation anomalies (colors). **g** The 200-hPa zonal wind anomalies (colors) from historical stratospheric nudging experiments. **h** As in (**g**) but for precipitation anomalies (colors). **i** East Asian

subtropical zonal wind anomalies (colors, averaged over 70–120°E, 15–27.5°N) lag-regressed on the standardized QBO index. NTA, SNA, and SIO indices are the weight-averaged SST anomalies over 0–80°W and 0–20°N, 0–60°W and 55–75°N, and 25–80°E and 35–50°S, respectively. The linear effects from NTA, SNA, and SIO have been removed in the anomalies in (**a–f, i**). Model results in (**g–h**) are averaged by 15 sets of historical stratospheric nudging experiments with different initial conditions, and each set of experiment runs for 25 years from 1995 to 2019. Dots indicate values at the 95% confidence level according to Student's t-test.

Here, $u$, $v$, and $T$ are the zonal wind, meridional wind, and temperature, respectively.

## Regression model

Predictions of precipitation, SPI, and their dipole mode index are obtained from multiple linear regression models in the following form:

$$Fcst = (a_1 \cdot QBO_{10-hPa} + a_2 \cdot QBO_{70-hPa}res) \\ + a_3 \cdot NTA + a_4 \cdot SNA + a_5 \cdot SIO + const \quad (3)$$

Here, $a_{1-5}$ are the regression coefficients. $Fcst$ represents the precipitation anomalies and the SPI at each grid, the DMI, and the DMI-SPI. NTA, SNA, SIO, and $QBO_{10-hPa}$ indicate the north tropical Atlantic, subpolar North Atlantic, and south Indian Ocean SST, and the 10-hPa QBO index, respectively. $QBO_{70-hPa}res$ is the 70-hPa QBO index with the linear portion related to the 10-hPa QBO removed. Note that both the 10-hPa and 70-hPa QBO index are used to adjust the downward propagation speed and improve forecasting skills. All predictors are chosen in March, 3 months earlier than the predicted values in July–August. The tolerance and variance inflation factor are shown in Supplementary Table 3.

Based on the above regression curves, the DMI or DMI-SPI time series can be established in the form:

$$DMI/DMI-SPI = (a_1 \cdot QBO_{10-hPa} + a_2 \cdot QBO_{70-hPa}res) \\ + a_3 \cdot NTA + a_4 \cdot SNA + a_5 \cdot SIO + \delta \quad (4)$$

where $a_{1-5}$, NTA, SNA, SIO, $QBO_{10-hPa}$, and $QBO_{70-hPa}res$ are the same as those in equation (3); $\delta$ is a residual term that the selected predictors cannot explain. The contributions of predictors to the DMI in 2020 and 2021 are as follows:

$$QBO_{2020,2021} = \frac{(a_1 \cdot QBO_{10-hPa} + a_2 \cdot QBO_{70-hPa}res)}{DMI} \quad (5)$$

$$NTA_{2020,2021} = \frac{a_3 \cdot NTA}{DMI} \quad (6)$$

$$SNA_{2020,2021} = \frac{a_4 \cdot SNA}{DMI} \quad (7)$$

$$SIO_{2020,2021} = \frac{a_5 \cdot SIO}{DMI} \quad (8)$$

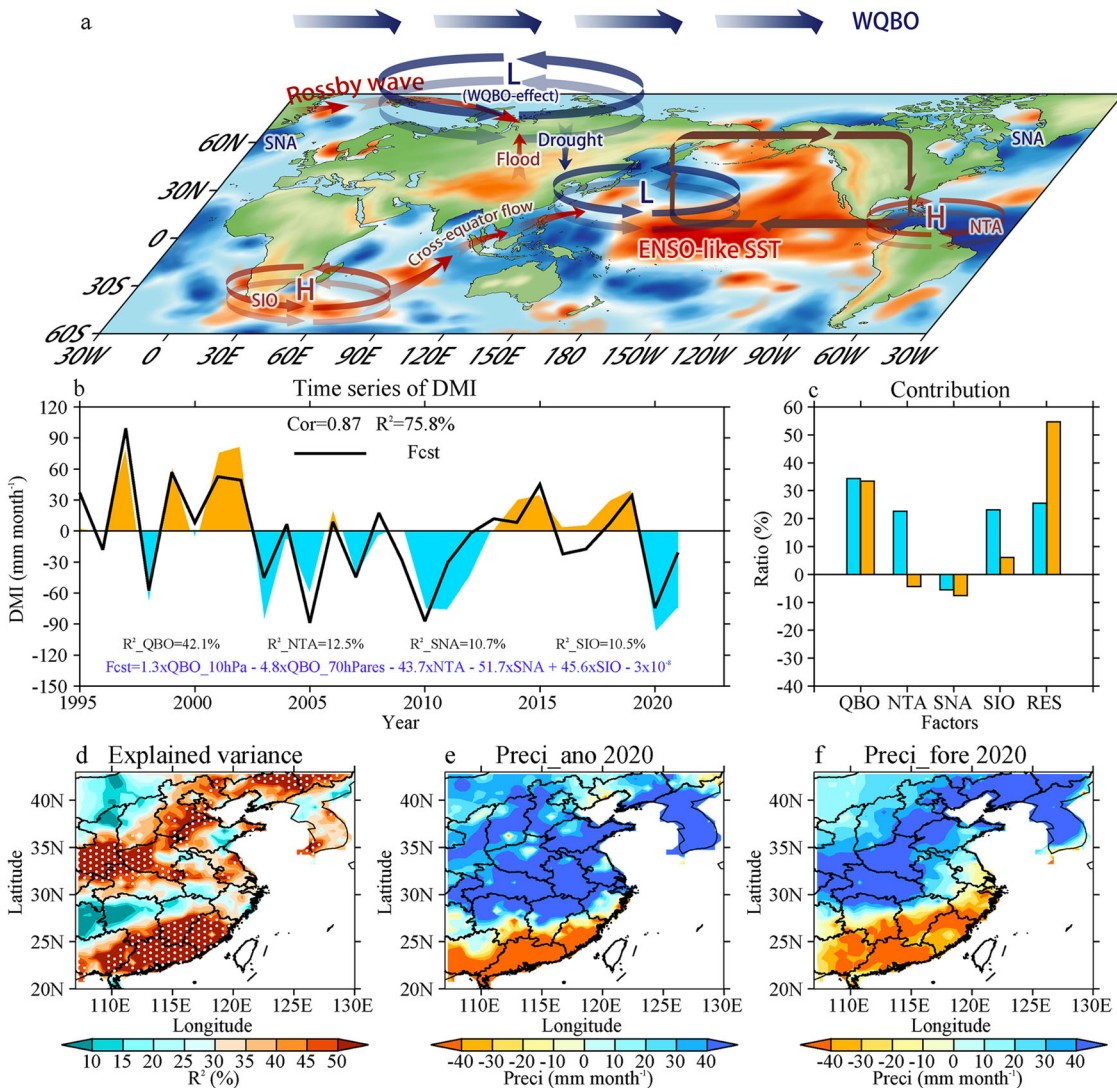

**Fig. 5 | Seasonal forecasting of July–August precipitation and dipole mode index (DMI) over East Asia. a** Schematic diagram for the factors impacting East Asia Summer Precipitation. **b** DMI (colors) and its forecasting curve. **c** Contributions of factors impacting the southern drought–northern flood (SDNF) pattern in 2020 and 2021. **d** Explained variance of the forecasting model at each grid. **e** Precipitation anomalies in 2020. **f** As in (**e**) but for forecasting results in 2020. H and L in (**a**) represent the abnormal high and abnormal low, respectively. A forecasting model is established based on the datasets over 1995–2019 to predict the precipitation at each grid and DMI anomalies in 2020 and 2021. NTA, SNA, and SIO indices are the weight-averaged SST anomalies over 0–80°W and 0–20°N, 0–60°W and 55–75°N, and 25–80°E, and 35–50°S, respectively. The NTA, SNA, and SIO, 10-hPa QBO index, and 70-hPa QBO index in March are chosen as the predictors, 3 months earlier than the predicted values. Note that both the 10-hPa and 70-hPa QBO index are used to fix the downward propagation speed and improve forecasting skills (see methods). Explained variances ($R^2$) of the different factors are shown in (**b**). The contributions in (**c**) are the ratio between the forecasted values and real precipitation (see methods). Dots in (**d**) indicate values at 95% confidence level according to *F*-test.

$$RES_{2020,2021} = \frac{\delta}{DMI} \tag{9}$$

## Rossby wave source

Based on previous study[65], the Rossby wave source (*RWS*) takes the following form:

$$RWS = -\nabla \cdot [V_\chi(\zeta + f)] = -\frac{\partial[u_\chi(\zeta + f)]}{\partial x} - \frac{\partial[v_\chi(\zeta + f)]}{\partial y} \tag{10}$$

where, $u_\chi$, $v_\chi$, $\zeta$ and $f$ are the divergent component of the zonal wind, divergent component of meridional wind, relative vorticity and Coriolis parameter, respectively. The first term in the right of equation (10) represents the zonal advection of absolute vorticity and zonal vortex stretching, while second term shows the meridional advection of absolute vorticity and meridional vortex stretching.

## Experimental design

Here, we use the Community Atmosphere Model version 6 (CAM6) in the Community Earth System Model version 2 (CESM2)[66] to verify QBO's effects (details in Supplementary Table 1). To avoid noise from the initial fields, we use different initial fields to drive the 15 sets of experiments. A control experiment (E0) is used to provide initial atmospheric states, based on climatological forcing fields such as climatological SST. Fifteen sets of nudging experiments are run to diagnose influences from the historical stratosphere; these are the whole stratospheric nudging experiments (E1). E1 has the same forcing fields as E0 to avoid the influence of ocean variations (such as ENSO), except for the stratosphere. The stratospheric conditions above the 143 hPa level in E1 are nudged to the JRA-55 reanalysis datasets with a

maximal nudging coefficient of 1.0. These nudging coefficients vary within a range of [0, 1.0], and a coefficient closer to 1 means the simulation is closer to the observations. The levels between 143 and 198 hPa are set as the transition layer, and no nudging is used below 198 hPa. This design guarantees a historical stratosphere in the experiments, avoiding influences from the lower top and poor performance of the stratosphere in CAM6. Each set of nudging experiments runs from March to August every year over the period of 1995-2019. Thus, an effect from the historical stratosphere can be observed in the modeling results.

## Data availability

The JRA-55 datasets are available online at http://jra.kishou.go.jp/. SST is obtained from the Met Office Hadley Centre (https://www.metoffice.gov.uk/hadobs/hadisst/). The station QBO index is obtained from the National Oceanic and Atmospheric Administration (NOAA, https://acd-ext.gsfc.nasa.gov/Data_services/met/qbo/qbo.html) of the United States. WNPSH datasets are from the China National Climate Centre (http://cmdp.ncc-cma.net/Monitoring/cn_stp_wpshp.php). Precipitation datasets are from the China National Meteorological Information Center (NMIC, https://data.cma.cn/) and Korea Meteorological Administration (KMA, https://data.kma.go.kr/data/grnd/selectAsosRltmList.do?pgmNo=36). CESM2-CAM6 sensitivity experiment outputs have already been uploaded[67].

## Code availability

The IDL was used for computations and plotting, available online at https://www.nv5geospatialsoftware.com/Support. The CESM2 model code are freely available from https://www.cesm.ucar.edu/models/cesm2. Codes used in this article for data analysis are available at the share repository[67].

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

## Acknowledgements

This research is jointly supported by the National Natural Science Foundation of China (Grants No. 42288101, 42120104001, 42192563, W.Z., 42130601, W.T., 42075060, J.L., 42005010, Y.Z., and 42275084, R.Z.), the Shanghai Sailing Program (23YF1401400, R.Z.) and the Natural Science Basic Research Program in Shaanxi Province of China (2022JM-142, R.Z.). We thank the scientific teams for the HadISST1, JRA-55, KMA, NMIC, and NOAA datasets. The authors also acknowledge NCAR for the CESM2 model.

## Author contributions

R.Z., W.Z., W.T., and J.L. developed the main idea and wrote the paper. R.Z. analyzed the observations and designed the experiments. Y.Z. analyzed the model results and J.Z. checked the station data.

## Competing interests

The authors declare no competing interests.
