## [Peer Review File · Nature Communications]

A stratospheric precursor of East Asian summer droughts and floodsEditorial Note: Parts of this Peer Review File have been redacted as indicated to remove third-party material where no permission to publish could be obtained.

REVIEWER COMMENTS

Reviewer #1 (Remarks to the Author):

Summary

This work tells an interesting story that stratospheric signal has a robust impact on the summer rainfall variability over East Asia, and can be used as a predictor. The authors reveal that the dominant mode of rainfall anomaly pattern over East Asia is characterized by a meridional dipole mode, with contrasting signs of anomalies to the south and north of 29/30°N. The authors show observational evidences on the possible impact by sea surface temperature (SST) anomalies and the stratospheric signal associated with QBO, and find that the stratospheric signal plays a critical role. Numerical experiments by nudging stratospheric circulation is adopted to verify the causal relationship and possible mechanism, and a statistical model is finally constructed to predict the precipitation anomaly over East Asia based on preceding stratospheric signal and SST anomalies. Overall, this work is well designed and the paper is clearly presented. I think this work can be accepted if the authors well address the following specific issues.

Specific Issues

1) In Fig. 2, the authors show the 5880 gpm contours for SFND and SDNF composites, and explain that a stronger subtropical high favors deficient rainfall over southern China and excessive rainfall over northern China. In fact, many previous studies suggested that it is misleading to measure the subtropical high based on a fixed contour (such as 5880 gpm) because geopotential height is proportional to local temperature and it rises substantially under global warming (e.g., Wu and Wang 2015; He et al. 2015, 2018). Also, previous studies also suggested that a stronger subtropical high may lead to excessive precipitation over southern China in summer (e.g., Yuan et al. 2017), and this contradiction with your result possibly results from the misleading effect of 5880 gpm contour. Therefore, I suggest showing the composite difference between SFND and SDNF cases in Fig. 2 for wind vectors and vertical velocity anomalies, instead of the 5880 gpm contour. In fact, the most important circulation anomaly for precipitation anomaly is local vertical velocity anomaly (Huang et al. 2013; Huang and Xie 2015), instead of geopotential height.

He C, Zhou T, Lin A, Wu B, Gu D, Li C, Zheng B (2015) Enhanced or Weakened Western North Pacific Subtropical High under Global Warming? *Scientific Reports* 5:16771. doi:10.1038/srep16771

Yuan Y, Gao H, Li W, Liu Y, Chen L, Zhou B, Ding Y (2017) The 2016 summer floods in China and associated physical mechanisms: A comparison with 1998. *J Meteorol Res* 31 (2):261-277. doi:10.1007/s13351-017-6192-5

He C, Lin A, Gu D, Li C, Zheng B, Wu B, Zhou T (2018) Using eddy geopotential height to measure the western North Pacific subtropical high in a warming climate. *Theor Appl Climatol* 131 (1):681-691. doi:10.1007/s00704-016-2001-9

Huang P, Xie S-P (2015) Mechanisms of change in ENSO-induced tropical Pacific rainfall variability in a warming climate. *Nature Geosci* 8 (12):922-926. doi:10.1038/ngeo2571

Huang P, Xie S-P, Hu K, Huang G, Huang R (2013) Patterns of the seasonal response of tropical rainfall to

global warming. *Nature Geosci* 6 (5):357-361. doi:10.1038/ngeo1792

2) L160-161: Why temperature advection anomaly stimulates rainfall anomaly? By stimulating vertical velocity anomaly via isentropic gliding? Some details are needed to make the mechanism clear to the readers. In addition, is the anomalous temperature advection dominated by the zonal or meridional wind anomaly? Some additional diagnosis is needed, in order to understand how the horizontal wind anomaly associated with QBO stimulates vertical velocity anomaly and precipitation anomaly.

3) It is interesting to note in Fig. 3a,b that the excessive (deficient) precipitation over southern (northern) part of East Asia is associated with local westerly (easterly) wind anomaly in the troposphere, but the linkage of zonal wind anomaly with precipitation is not very clearly explained. I guess a possible mechanism is that there is a warm center over Tibetan Plateau, so that westerly (easterly) wind anomaly may stimulates ascent (descent) on the eastern side of Tibetan Plateau over East Asia via warm (cold) advection (Sampe and Xie 2010; He et al. 2023).

Sampe T, Xie SP (2010) Large-Scale Dynamics of the Meiyu-Baiu Rainband: Environmental Forcing by the Westerly Jet. *J Climate* 23 (1):113-134. doi:10.1175/2009jcli3128.1

He C, Zhou T, Zhang L, Chen X, Zhang W (2023) Extremely hot East Asia and flooding western South Asia in the summer of 2022 tied to reversed flow over Tibetan Plateau. *Clim Dynam*. doi:10.1007/s00382-023-06669-y

4) In the CAM6 nudging experiment, is the nudging of stratospheric wind anomaly applied globally or confined within the tropics? In addition, some detailed description is needed to explain how the stratospheric signal propagates into the troposphere.

5) The QBO signal is a rather zonal uniform signal. Over other longitudinal bands besides East Asia, does QBO also stimulates similar meridional dipoles of zonal wind anomaly and precipitation anomaly between the northern and southern sides of 29/30°N? If not, why is East Asia so special?

6) The percentage variance is stated as 72% in the abstract (L11) but 75% (L188) in the main text. Which is correct?

7) Fig. 4 shows two cases for the summers of 2020 and 2021. The observed precipitation anomalies show very similar pattern between 2020 and 2021. I think it may be better to show two contrasting cases, or only one case together with an spatial pattern plot for the overall hindcast skill based on all available data.

Reviewer #2 (Remarks to the Author):

In this study, the authors state that the QBO can influence precipitation over East Asia through

modulation of tropospheric zonal winds. My major concern is the downward propagation of the QBO signal that can alter the wind anomalies over the subtropics, and the generation of the negative wind anomalies over the midlatitude. Besides, the authors talk too much about the influence of the SST anomalies over the Atlantic and the southern Indian Ocean, which is not closely linked to the QBO influence on the precipitation over East Asia. The authors should reorganize the manuscript to focus on the physical process that QBO can influence the precipitation over East Asia in the revision. This manuscript needs major revision before considering to be published in Nature Communications.

Major comments:

- 1) The physical processes about the downward propagation of the QBO signal that can alter the wind anomalies over the subtropics, and the generation of the negative wind anomalies over the midlatitude are not convincing;
- 2) The calculation of wave activity flux should be checked. What kind of wave flux did the author use in this study?

Specific comments:

Lines 54-59: The authors should give a more specific review on the relationship between the QBO and East Asian climate;

Lines 61-62: Why do you focus on the mechanisms after 1995?

Lines 85-88: The QBO index should be plotted in Figure 1d;

Lines 110-111: These signs are too small to be distinguished;

Lines 121-122: The longitudes and latitudes of the NTA, SNA and SIO regions should be marked in Figures 2b-d;

Figure 3b: The wave flux vectors are too small;

Lines 144-148: It is not Kelvin wave that drives the acceleration of the zonal winds. The authors should present the EP flux and its divergence when talking about the acceleration of the zonal winds. The downward propagation of the QBO signal should be shown in a temporal evolution map.

Figure 5c: Please check the calculation of the wave flux;

Lines 151-153: Why did the temperature gradient anomalies can extend the QBO wind signal to the troposphere?

Lines 153-156: What kind of wave-flow interaction can influence the wind anomalies over the subtropics and mid-latitude? The authors should present more detailed information on the generation of the negative wind anomalies over the mid-latitude. The current statement is not convincing. Furthermore, please check the calculation of the wave flux in Figure 3c;

Lines 158-160: It is barotropic structure that extends from the upper troposphere to the middle troposphere, not downward extension;

Lines 160-162: Please rewrite this ambiguous sentence.

Reviewer #3 (Remarks to the Author):

This work analyzed a stratospheric precursor of East Asian summer droughts and floods, which is the QBO. The authors indicated that the QBO could modulate the East Asian precipitation via a subtropical way, contributing the largest explained variation of the south-north dipole pattern of summer rainfall in East Asia. This is an interesting and important topic, which provides a potential prediction signal for seasonal prediction in China. Overall, the manuscript is well written, and the results are credible. The contents appear suitable for publication, but some necessary revision is needed.

1. In the “Methods” section, it is provided how to establish a multiple linear equation associated with the dipole mode of summer rainfall, but each coefficient of the final linear equation is not provided established through historical data. This should be one of the most important results of this work, so it is suggested to be clear, which is also an indicator that intuitively reflects the contribution of various variables. In addition, the manuscript did not consider whether the selected indicators are independent of each other, which should also be explained.

2. The manuscript specifically evaluated the contributions of QBO in 2020 and 2021, and it is suggested that the specific values of the QBO index in 2020 and 2021 should be provided in the manuscript, or the time series of the QBO index is extended to 2022 in Fig. S8.

3. Readers may be curious that since QBO is quasi-biennial oscillation signal, and the summer rainfall in 2020 and 2021 was significantly affected by QBO, why do the July-August rainfall anomalies in 2020 and 2021 exhibited a consistent spatial distribution of SDNF mode? The manuscript also mentioned that QBO has an impact on the atypical north-south reverse mode in 2022, and how did this affect it? It is suggested to add more discussions on these issues in the “Discussion” section.

4. In the “Data” section, the materials and climate models used should have corresponding references. For example, the West Pacific subtropical high indices can be supplemented with the following references:

Reconstruction and application of the monthly western Pacific subtropical high indices (in Chinese). *J. Appl. Meteor. Sci.*, 2021, 23: 414-423.

The Asian Summer Monsoon: Characteristics, Variability, Teleconnections and Projection. 2019, pp 1-237. doi: 10.1016/C2017-0-04074-0. Elsevier.

**REVIEWER COMMENTS**

Reviewer #1 (Remarks to the Author):

Summary

This work tells an interesting story that stratospheric signal has a robust impact on the
summer rainfall variability over East Asia, and can be used as a predictor. The authors
reveal that the dominant mode of rainfall anomaly pattern over East Asia is
characterized by a meridional dipole mode, with contrasting signs of anomalies to the
south and north of 29/30°N. The authors show observational evidences on the possible
impact by sea surface temperature (SST) anomalies and the stratospheric signal
associated with QBO, and find that the stratospheric signal plays a critical role.
Numerical experiments by nudging stratospheric circulation is adopted to verify the
causal relationship and possible mechanism, and a statistical model is finally
constructed to predict the precipitation anomaly over East Asia based on preceding
stratospheric signal and SST anomalies. Overall, this work is well designed and the
paper is clearly presented. I think this work can be accepted if the authors well
address the following specific issues.

**Response: Thank you for the comments. We have added more information in the**
**Influence of sea surface temperature and Role of the Quasi-Biennial Oscillation**
**section, we also highlight the main changes in the revised manuscript.**

Specific Issues

1) In Fig. 2, the authors show the 5880 gpm contours for SFND and SDFN
composites, and explain that a stronger subtropical high favors deficient rainfall over
southern China and excessive rainfall over northern China. In fact, many previous
studies suggested that it is misleading to measure the subtropical high based on a
fixed contour (such as 5880 gpm) because geopotential height is proportional to local
temperature and it rises substantially under global warming (e.g., Wu and Wang 2015;
He et al. 2015, 2018). Also, previous studies also suggested that a stronger subtropical
high may lead to excessive precipitation over southern China in summer (e.g., Yuan et
al. 2017), and this contradiction with your result possibly results from the misleading
effect of 5880 gpm contour. Therefore, I suggest showing the composite difference

between SFND and SDNF cases in Fig. 2 for wind vectors and vertical velocity
anomalies, instead of the 5880 gpm contour. In fact, the most important circulation
anomaly for precipitation anomaly is local vertical velocity anomaly (Huang et al.
2013; Huang and Xie 2015), instead of geopotential height.

**Response: Thank you for pointing this out. We reorganized Figure 2 and added**
**the vertical velocity and moisture flux anomalies (Figure R1). The precipitation**
**is related to both vertical velocity and water vapor transport, according to the**
**water-vapor budget equation: $P \approx -(\omega \partial_p q) - (V_h \cdot \nabla_h q) + E + residual - q$, where P ,**
**q , V , E , and ω are the precipitation, water vapor, horizontal wind, evaporation,**
**and omega. As you said, there is anomalous ascending motion in the southern**
**region of East Asia and the water vapor flux converges there (Fig R1c), which**
**favors precipitation. Meanwhile, the anomalous sinking motion and the**
**divergence of water vapor suppress the precipitation in the northern region of**
**East Asia. The anomalous vertical motion is related to temperature advection**
**based on the omega equation; that is, ascending (sinking) motion tends to occur**
**in a warm (cold) temperature advection region. Figure R2 (Supplementary Fig.2**
**in the revised manuscript) shows the composite differences (SFND minus SDNF)**
**in the temperature advection anomalies. We can see the anomalous warm (cold)**
**temperature advection in the southern (northern) region of East Asia (Fig. R2a),**
**corresponding to the vertical motion in Fig. R1b. Both the zonal term and**
**meridional term make positive contributions to the temperature advection**
**anomalies (Fig. R2b-c). For the moisture flux, meridional water vapor transport**
**plays the most important role (Fig. R2d-f). This information has been added in**
**the revised manuscript (see Fig. 2 and Supplementary Fig. 2).**

He C, Zhou T, Lin A, Wu B, Gu D, Li C, Zheng B (2015) Enhanced or Weakened Western North
Pacific Subtropical High under Global Warming? Scientific Reports 5:16771.
doi:10.1038/srep16771

Yuan Y, Gao H, Li W, Liu Y, Chen L, Zhou B, Ding Y (2017) The 2016 summer floods in China
and associated physical mechanisms: A comparison with 1998. J Meteorol Res 31 (2):261-277.
doi:10.1007/s13351-017-6192-5

He C, Lin A, Gu D, Li C, Zheng B, Wu B, Zhou T (2018) Using eddy geopotential height to
measure the western North Pacific subtropical high in a warming climate. Theor Appl Climatol

131 (1):681-691. doi:10.1007/s00704-016-2001-9
 Huang P, Xie S-P (2015) Mechanisms of change in ENSO-induced tropical Pacific rainfall
 variability in a warming climate. Nature Geosci 8 (12):922-926. doi:10.1038/ngeo2571
 Huang P, Xie S-P, Hu K, Huang G, Huang R (2013) Patterns of the seasonal response of tropical
 rainfall to global warming. Nature Geosci 6 (5):357-361. doi:10.1038/ngeo1792

 **Figure R1 (Figure 2 in the revised manuscript). Linkages between the DMI and detrended**
 **SST/circulation anomalies over the period 1995-2019. a** Simultaneous correlation in the global
 ocean. **b** Composite differences (SFND minus SDFN) of horizontal circulation (vectors) and
 vertical velocity (colors) anomalies at 500 hPa. **c** As in **b** but for vertically integrated moisture flux
 (vectors) and its divergence (colors) anomalies. **d** Lag correlation over the Atlantic region
 (averaged over 0-80°W). **e** Lag correlation over the tropical region (averaged over 10°S-10°N). **f**
 Lag correlation over the Indian Ocean (averaged over 30-60°E). The locations of the western
 points of the WNPSH ridge and strength of the WNPSH are indicated by triangles and plus signs
 in **a**, respectively. Orange/black triangles and plus signs represent the SDFN/ SFND years and a
 larger plus sign means a stronger WNPSH. The NTA, SNA, and SIO indices are the
 weight-averaged SST anomalies over 0-80°W & 0-20°N, 0-60°W & 55-75°N, and 25-80°E &
 35-50°S, respectively. Dashed boxes in **d-f** show the locations of NTA, SNA, and SIO. Straight
 lines in **d-f** are July and August, and dots indicate values at the 95% confidence level.

 **Figure R2 (Supplementary Fig. 2 in the revised manuscript) Composite differences (SFND**
 **minus SDNF) of circulation anomalies. a** Horizontal wind (vectors) and temperature advection
 (colors) anomalies at 500 hPa. **b** Horizontal wind (vectors) and zonal term of temperature
 advection (colors) and anomalies at 500 hPa. **c** Meridional term of temperature advection (colors)
 and horizontal wind (vectors) anomalies at 500 hPa. **d** Vertically integrated moisture flux (vectors)
 and its divergence (colors) anomalies. **e** Vertically integrated moisture flux (vectors) and zonal
 term of its divergence (colors) anomalies. **f** Vertically integrated moisture flux (vectors) and
 meridional term of its divergence (colors) anomalies. Dots indicate values at the 95% confidence
 level.

2) L160-161: Why temperature advection anomaly stimulates rainfall anomaly? By
 stimulating vertical velocity anomaly via isentropic gliding? Some details are needed
 to make the mechanism clear to the readers. In addition, is the anomalous temperature
 advection dominated by the zonal or meridional wind anomaly? Some additional
 diagnosis is needed, in order to understand how the horizontal wind anomaly
 associated with QBO stimulates vertical velocity anomaly and precipitation anomaly.

**Response: Thanks. QBO can influence precipitation by modulating temperature**
 **advection and water vapor transport. According to the geostrophic adjustment**
 **and the thermal wind theory, QBO-induced zonal wind and wave-flow feedback**
 **lead to an abnormal low and cold center over East Asia. This abnormal low and**
 **its corresponding temperature fields can extend from the upper atmosphere to**

the lower atmosphere (Fig. R3a-b). Then, the horizontal wind transports more
 warm air into the southern region of East Asia and cold air into the northern
 region of East Asia, controlling the vertical motion there (Fig. R3c-d, R4a-c).
 Both the zonal term and the meridional term play a positive role in the
 temperature advection, and the zonal term is larger than the meridional term
 (Fig. R4a-c). In addition, the transport of water vapor also plays an important
 role in the QBO-induced precipitation. Anomalous water vapor flux
 convergence/divergence leads to more/less precipitation over the
 southern/northern region of East Asia (Fig. R3e-f, R4d-f). This information has
 been added in the revised manuscript (see Role of the Quasi-Biennial Oscillation
 section).

**Figure R3 (Fig. 4 in the revised manuscript). East Asian detrended circulation**
 **anomalies regressed on the 30-hPa standardized QBO index over the period**
 **1995-2019.** a East Asian geopotential height anomalies (colors, averaged over
 70-120°E). b As in a but for the temperature anomalies (colors). c The 500-hPa
 vertical velocity anomalies (colors). d Vertical velocity anomalies averaged over
 105-120°E (colors). e Vertically integrated moisture flux (vectors) and its divergence
 (colors) anomalies. f Observational precipitation anomalies (colors). The linear effects
 from NTA, SNA, and SIO have been removed in the anomalies in a-f, i. Dots indicate
 values at the 95% confidence level.

**Figure R4 (Supplementary Fig. 7) Temperature advection and moisture flux anomalies**
 **regressed on the standardized 30-hPa QBO index. a** Temperature advection anomalies (colors)
 **at 500 hPa. b** Zonal term of temperature advection anomalies (colors) at 500 hPa. **c** Meridional

3) It is interesting to note in Fig. 3a,b that the excessive (deficient) precipitation over southern (northern) part of East Asia is associated with local westerly (easterly) wind anomaly in the troposphere, but the linkage of zonal wind anomaly with precipitation is not very clearly explained. I guess a possible mechanism is that there is a warm center over Tibetan Plateau, so that westerly (easterly) wind anomaly may stimulates ascent (descent) on the eastern side of Tibetan Plateau over East Asia via warm (cold) advection (Sampe and Xie 2010; He et al. 2023).

Response: Thanks. We have rechecked the temperature anomaly here. There is an anomalous cold center over East Asia, including the Tibetan Plateau, rather than a warm center (Fig. R3b). The temperature advection anomalies lead to anomalous vertical motion over East Asia, which further influences the

148 precipitation. The zonal wind anomaly contributes only part of the temperature
 advection anomalies, and temperature advection anomalies are the combined
 result of zonal wind, meridional wind, and temperature anomalies (Figure R5).
 To better understand this process, we divided the variables into two parts, the
 climatological values and the deviations from the climatological values. Then,
 temperature advection anomalies can be rewritten as the following terms:

$$\begin{aligned}
 & -(V \nabla T)^a = -(V^c \nabla T^a + V^a \nabla T^c + V^a \nabla T^a) = \\
 & -(u^c T_x^a + u^a T_x^c + u^a T_x^a + v^c T_y^a + v^a T_y^c + v^a T_y^a) \\
 & = -\underbrace{(u^c T_x^a + u^a T_x^c + v^c T_y^a + v^a T_y^c)}_{\text{Linear term}} - \underbrace{(u^a T_x^a + v^a T_y^a)}_{\text{Nonlinear term}}
 \end{aligned}$$

 Where u^c , v^c , and T^c are the climatological values of zonal wind, meridional wind,
 and temperature, respectively, and u^a , v^a , and T^a are the zonal wind, meridional
 wind, and temperature anomalies, respectively.

We can see that the linear term plays a dominant role in the temperature
 advection anomalies (Fig. R5a-b), whereas the nonlinear term makes a negative
 contribution (Fig. R5a, c). The cold advection anomalies in the northern region
 of East Asia are related to the $-u^c \cdot T_x^a$ and $-v^a \cdot T_y^c$ terms (Fig. R5d, h), and warm
 advection anomalies in the southern region of East Asia are related to the $-u^a \cdot T_x^c$
 and $-v^c \cdot T_y^a$ terms (Fig. R5e, g). Some terms lack statistical significance; thus, we don't
 add them in the revised manuscript. In general, the temperature advection
 anomalies are the combined result of zonal wind, meridional wind, and
 temperature anomalies.

Sampe T, Xie SP (2010) Large-Scale Dynamics of the Meiyu-Baiu Rainband: Environmental
 Forcing by the Westerly Jet. J Climate 23 (1):113-134. doi:10.1175/2009jcli3128.1

He C, Zhou T, Zhang L, Chen X, Zhang W (2023) Extremely hot East Asia and flooding western
 South Asia in the summer of 2022 tied to reversed flow over Tibetan Plateau. Clim Dynam.
 doi:10.1007/s00382-023-06669-y

**Figure R5. Temperature advection anomalies at 500 hPa regressed on the standardized**
 **30-hPa QBO index. a** The temperature advection anomalies. **b** The linear term. **c** The nonlinear
 term. **d** $-u^c \cdot T_x^a$ term. **e** $-u^a \cdot T_x^c$ term. **f** $-u^a \cdot T_x^a$ term. **g** $-v^c \cdot T_y^a$ term. **h** $-v^a \cdot T_y^c$ term. **i** $-v^a \cdot T_y^a$ term. Dots
 indicate values at the 90% confidence level.

4) In the CAM6 nudging experiment, is the nudging of stratospheric wind anomaly
 applied globally or confined within the tropics? In addition, some detailed description
 is needed to explain how the stratospheric signal propagates into the troposphere.

**Response; Thanks. It is global nudging, because CAM6 cannot rebuild a real**
 **stratospheric condition in the mid- to high latitudes. For instance, it has some**
 **biases in the eddy heat flux in the mid- to high stratosphere and cannot rebuild a**
 **real QBO. To reduce the noise from the mid- to high-latitude stratosphere, we**
 **use global nudging.**

**The downward propagation of stratospheric signals includes two processes:**
 **QBO-induced secondary circulation and tropospheric wave-flow feedback.**
 **Figure R6 shows the QBO-induced circulation anomalies over East Asia. We can**
 **see that the QBO-induced zonal wind pattern is similar to the DMI (Fig. R6a-b).**
 **Previous studies have shown that the tropical Kelvin wave and gravity wave**
 **force the easterly and westerly wind to propagate downward, generating a**

quasi-biennial oscillation. To maintain the momentum and thermal wind balance,
QBO induces secondary circulations from the tropical upper troposphere to the
stratosphere. In the Northern Hemisphere, a negative stream function anomaly is
observed near 30 hPa, accompanied by two positive stream function anomalies
on its upper and lower sides (Fig. R6c). Figure R6d shows the development of
these stream function centers; the positive and negative centers propagate
downward from the upper stratosphere into the troposphere, following the QBO
signals. In the Northern Hemisphere, a positive/negative stream function means
an anticyclonic (clockwise)/cyclonic (anticlockwise) circulation. According to the
relationship between the zonal wind and stream function ($u = -(\partial\psi / \partial y)$),
westerly anomalies are located on the north side of positive stream function
anomalies. Thus, with the positive stream function anomalies appearing in the
tropical upper troposphere (Fig. R6c-d), westerly anomalies occur in the
subtropical troposphere (Fig. R6b). The westerly anomalies disturb the Rossby
wave sources (RWS), leading to a positive RWS anomaly in the subtropical
region and poleward Rossby wave flux anomalies (Fig. R6e). The Rossby wave
anomalously diverges in the subtropical region and converges in the
mid-latitudes, amplifying the subtropical westerly anomalies and forcing
mid-latitude easterly anomalies (Fig. R6f). These zonal wind anomalies can result
in a cyclonic anomaly over East Asia (contours, Fig. R6f) and further influence
the precipitation. A detailed description has been added to the revised
manuscript (see Role of the Quasi-Biennial Oscillation section).

 **Figure R6. (Fig. 3). Detrended circulation anomalies regressed on the DMI and 30-hPa QBO**
 **index over the period 1995-2019. a** East Asian zonal wind anomalies (colors, averaged over
 70-120°E) regressed on the standardized DMI index. **b** East Asian zonal wind (colors) and Plumb
 wave flux (vectors) anomalies averaged over 70-120°E regressed on the standardized QBO index.
 **c** Zonal mean stream function anomalies (colors) regressed on the standardized QBO index. **d**
 Tropical zonal mean stream function anomalies (colors, averaged over 0-15°N) lag-regressed on
 the standardized QBO index. **e** The 200-hPa Rossby wave source (colors) and Plumb wave flux
 (vectors) anomalies regressed on the standardized QBO index. **f** The 200-hPa zonal wind (colors)
 and geopotential height (contours) anomalies regressed on the standardized QBO index. The units
 of geopotential height are gpm in **f**. Linear effects from NTA, SNA, and SIO have been removed
 in the anomalies in **b-f**. Dots indicate values at the 95% confidence level.

5) The QBO signal is a rather zonal uniform signal. Over other longitudinal bands
 besides East Asia, does QBO also stimulates similar meridional dipoles of zonal wind
 anomaly and precipitation anomaly between the northern and southern sides of
 29/30°N? If not, why is East Asia so special?

**Response: The downward extension of zonal wind anomalies in the subtropics is**
 **a direct effect of QBO-induced meridional circulation. According to a previous**
 **study, meridional dipoles often occur near the exit of the subtropical jet,**
 **depending on the positive wave-flow feedback there. Figure R7d-f shows the**
 **location of the subtropical jet and the response to EQBO at 297 hPa. We can see**

that the easterly wind response (Fig. R7e-f) is strong and significant near the exit
of the subtropical jet (Fig. R7d). Considering the location of the subtropical jet in
the Northern Hemisphere, we can observe such meridional dipoles in the East
Asia–North Pacific or North Atlantic–Europe regions (Figure R8).

[REDACTED]

Figure R7. The mean zonal wind and response to EQBO (from Garfinkel et al. 2011, Fig. 12). (d)
Mean zonal wind in the control case. (e) Response to EQBO at the 297-hPa level in days 45–75 (f)
As in (e) but for days 70–120. Stars mark the jet maximum at each longitude. Negative contours
are thick, and significant regions at 95% are shaded in (e-f).

Garfinkel, C. I., and D. L. Hartmann, 2011: The Influence of the Quasi-Biennial Oscillation
on the Troposphere in Winter in a Hierarchy of Models. Part I: Simplified Dry GCMs. *J Atmos Sci*,
68, 1273-1289.

**Figure R8.** a Zonal wind anomalies over the Atlantic-Europe region (30W-30E) regressed on the
30-hPa QBO index. b As in a but for East Asia (70-120E).

6) The percentage variance is stated as 72% in the abstract (L11) but 75% (L188) in
the main text. Which is correct?

**Response: Thanks. We updated the station datasets, with more stations to**
**estimate missing data and do a homogeneity test. Thus, the variances have little**
**change. They are not the same explained variance. As shown in Figure R9, the**
**73.1% is for the SPI index and the 75.8% is for the precipitation anomalies. We**
**have revised the abstract and retain only the 75.8%.**

**Figure R9, a** DMI (colors) and its forecasting curve (black line). **b** DMI_SPI (colors) and its
 forecasting curve (black line).

7) Fig. 4 shows two cases for the summers of 2020 and 2021. The observed
 precipitation anomalies show very similar pattern between 2020 and 2021. I think it
 may be better to show two contrasting cases, or only one case together with an spatial
 pattern plot for the overall hindcast skill based on all available data.

**Response: Thanks. We have revised this figure. There are not two contrasting**
 **cases in the last several years, so we added a spatial pattern plot for the**
 **explained variance (Figure R10).**

**Fig. R10 (Fig. 5d-f in the revised manuscript). Seasonal forecasting of July-August**
 **precipitation and DMI over East Asia. d** Explained variance of the forecasting model at each
 **grid. e** Precipitation anomalies in 2020. **f** As in **e** but for forecasting results in 2020.

Reviewer #2 (Remarks to the Author):

In this study, the authors state that the QBO can influence precipitation over East Asia
through modulation of tropospheric zonal winds. My major concern is the downward
propagation of the QBO signal that can alter the wind anomalies over the subtropics,
and the generation of the negative wind anomalies over the midlatitude. Besides, the
authors talk too much about the influence of the SST anomalies over the Atlantic and
the southern Indian Ocean, which is not closely linked to the QBO influence on the
precipitation over East Asia. The authors should reorganize the manuscript to focus on
the physical process that QBO can influence the precipitation over East Asia in the
revision. This manuscript needs major revision before considering to be published in
Nature Communications.

**Response: Thank you for the comments. The physical mechanism is the most**
**important process in this study. We have added more information about the**
**QBO-induced process in the Role of the Quasi-Biennial Oscillation section, we**
**also highlight the main changes in the revised manuscript.**

Major comments:

1) The physical processes about the downward propagation of the QBO signal that
can alter the wind anomalies over the subtropics, and the generation of the negative
wind anomalies over the midlatitude are not convincing;

**Response: Thanks. We have given more information about the physical**
**mechanism. Usually, there are three ways in which QBO influences the surface:**
**the polar way impacting the stratospheric polar vortex, the tropical way**
**impacting tropical static stability and convective activity, and the subtropical**
**way modulating the subtropical jet and wave activity. In the summer hemisphere,**
**there is no stratospheric polar vortex, and the tropical response in our study**
**lacks statistical significance. Thus, we focus mainly on the subtropical way here.**
**According to a classical study (Garfinkel et al. 2011), the subtropical way**
**involves two key processes: QBO-induced secondary circulation (QBO's direct**

effect) and tropospheric wave-flow feedback (enhancing tropospheric QBO
signals). Figure R1 shows the zonal wind responses to EQBO and 3xEQBO
**without wave-flow feedback**. We can see that the EQBO-induced easterly wind
can directly extend to the subtropical upper troposphere (Fig. R1d), and a
stronger QBO will cause stronger responses (Fig. R1e). However, we cannot see a
strong westerly response in the mid-latitudes in the case without wave-flow
feedback. Figure R2 shows the zonal wind responses to EQBO and 3xEQBO **with**
**wave-flow feedback**. The stronger easterly wind can extend to the surface in the
subtropics, and obvious westerly responses can be observed in the mid-latitude
troposphere. Thus, it is the tropospheric wave-flow feedback that enhances
QBO's effect. The responses of tropospheric zonal wind are the results of
QBO-induced direct circulation and its corresponding wave-flow feedback in the
subtropical and mid-latitude. Figure R3 further shows the horizontal distribution
of mean zonal wind and the responses to EQBO. In both the EQBO and
3xEQBO experiments, the strong easterly responses occur mainly in the exit of
the subtropical jet (Fig. R3b-c, e-f). The response to WQBO is similar to the
response to EQBO, but with opposite sign. This is why the QBO's subtropical
effect is strong over East Asia.

The physical processes in our study are shown in Figures R4-5. Figure R4
shows the QBO-induced circulation anomalies over East Asia. We can see that
the QBO-induced zonal wind pattern is similar to the DMI (Fig. R4a-b). The
tropical Kelvin wave and gravity wave force the tropical stratospheric easterly
and westerly wind to propagate downward, generating a quasi-biennial
oscillation. To maintain the momentum and thermal wind balance, QBO induces
secondary circulations from the tropical upper troposphere to the upper
stratosphere. Then, a negative stream function anomaly is observed near 30 hPa
in the Northern Hemisphere, accompanied by two positive stream function
anomalies on its upper and lower sides (Fig. R4c). These positive and negative
centers propagate downward from the upper stratosphere into the upper
troposphere (Fig. R4d), following the downward propagation of QBO. In the

Northern Hemisphere, a positive/negative stream function means an anticyclonic
(clockwise)/cyclonic (anticlockwise) circulation. According to the relationship
between the zonal wind and stream function ($u = -(\partial\psi/\partial y)$), westerly
anomalies are located on the north side of positive stream function anomalies.
Thus, with the positive stream function anomalies appearing in the tropical
upper troposphere (Fig. R4c-d), westerly anomalies occur in the subtropical
troposphere (Fig. 4b). These westerly anomalies disturb the Rossby wave sources
(RWS), leading to a positive RWS anomaly in the subtropics and Rossby wave
anomalies propagating poleward. The Rossby wave anomalously diverges in the
subtropics and converges in the mid-latitudes (Fig. R4e), amplifying the
subtropical westerly anomalies and forcing mid-latitude easterly anomalies (Fig.
R4f). Based on the geostrophic adjustment and thermal wind theory, the zonal
wind anomalies can result in a cyclonic anomaly over East Asia (contours, Fig.
R4f). This cyclonic anomaly extends from the stratosphere to the surface,
accompanied by a cold center in the troposphere (Fig. R5a-b). The relevant
horizontal wind and temperature anomalies lead to changes in temperature
advection and further influence the vertical motion (Fig R5c-d). Anomalous
circulation also influences water vapor flux and finally precipitation (Fig R5e-f).
A more detailed description has been added to the revised manuscript (Role of
the Quasi-Biennial Oscillation), and we rewrote this section.

[REDACTED]

Figure R1 Zonal wind responses to the EQBO **without wave-flow feedback**. d Forced by

EQBO e Forced by Triple EQBO. Thick lines represent easterly wind and thin lines are westerly
wind. From Garfinkel et al. 2011.

[REDACTED]

**Figure R2 Zonal wind responses to the EQBO with wave-flow feedback.** f Forced by EQBO. h
Forced by Triple EQBO. Thick lines represent easterly wind and thin lines are westerly wind.
From Garfinkel et al. 2011.

[REDACTED]

**Figure R3. (a), (d) Mean zonal wind in the control case and (b),(c),(e),(f) response to**
**EQBO—(middle) days 45–75 and (right) days 70–120 at the 297-hPa level.** For (a),(d), regions
with zonal wind above 30 m/s are shaded light gray. For (b),(c),(e),(f), negative contours are thick;
significant regions at 95% are shaded; and stars mark the jet maximum at each longitude. From
Garfinkel et al. 2011.
Ref: Garfinkel, C. I., and D. L. Hartmann, 2011: The Influence of the Quasi-Biennial Oscillation
on the Troposphere in Winter in a Hierarchy of Models. Part I: Simplified Dry GCMs. J Atmos Sci,
68, 1273-1289.

 **Figure R4 (Fig. 3 in the revised manuscript). Detrended circulation anomalies regressed on**
 **the DMI and 30-hPa QBO index over the period 1995-2019. a** East Asian zonal wind
 wind anomalies (colors, averaged over 70-120°E) regressed on the standardized DMI index. **b** East
 Asian zonal wind (colors) and Plumb wave flux (vectors) anomalies averaged over 70-120°E
 regressed on the standardized QBO index. **c** Zonal mean stream function anomalies (colors)
 regressed on the standardized QBO index. **d** Tropical zonal mean stream function anomalies
 (colors, averaged over 0-15°N) lag-regressed on the standardized QBO index. **e** The 200-hPa
 Rossby wave source (colors) and Plumb wave flux (vectors) anomalies regressed on the
 standardized QBO index. **f** The 200-hPa zonal wind (colors) and geopotential height (contours)
 anomalies regressed on the standardized QBO index. The units of geopotential height are gpm in **f**.
 Linear effects from NTA, SNA, and SIO have been removed in the anomalies in **b-f**. Dots indicate
 values at the 95% confidence level.

**Figure R5 (Fig. 4 in the revised manuscript). East Asian detrended circulation anomalies**
 **regressed on the 30-hPa standardized QBO index over the period 1995-2019. a** East Asian
 **geopotential height anomalies (colors, averaged over 70-120°E). b** As in **a** but for the temperature
 **anomalies (colors). c** The 500-hPa vertical velocity anomalies (colors). **d** Vertical velocity
 **anomalies averaged over 105-120°E (colors). e** Vertically integrated moisture flux (vectors) and its
 **divergence (colors) anomalies. f** Observational precipitation anomalies (colors). **g** The 200-hPa
 **zonal wind anomalies (colors) from real stratospheric nudging experiments. h** As in **g** but for
 **precipitation anomalies (colors). i** East Asian subtropical zonal wind anomalies (colors, averaged
 **over 70-120°E, 15-27.5°N) lag-regressed on the standardized QBO index. The linear effects from**
 **NTA, SNA, and SIO have been removed in the anomalies in a-f, i. Model results in g-h** are
 **averaged by 15 sets of real stratospheric nudging experiments with different initial conditions, and**
 **each set of experiments runs for 25 years from 1995 to 2019. Dots indicate values at the 95%**
 **confidence level.**

2) The calculation of wave activity flux should be checked. What kind of wave flux
 did the author use in this study?

Response: It is the Plumb wave activity flux as follows:

$$F_s = p \cos \varphi \left(\begin{array}{l} \frac{1}{2a^2 \cos^2 \varphi} \left[\left(\frac{\partial \psi'}{\partial \lambda} \right)^2 - \psi' \frac{\partial^2 \psi'}{\partial \lambda^2} \right] \\ \frac{1}{2a^2 \cos^2 \varphi} \left(\frac{\partial \psi'}{\partial \lambda} \frac{\partial \psi'}{\partial \varphi} - \psi' \frac{\partial^2 \psi'}{\partial \lambda \partial \varphi} \right) \\ \frac{2\Omega^2 \sin^2 \varphi}{N^2 a \cos \varphi} \left(\frac{\partial \psi'}{\partial \lambda} \frac{\partial \psi'}{\partial z} - \psi' \frac{\partial^2 \psi'}{\partial \lambda \partial z} \right) \end{array} \right)$$

Where $\psi = \Phi / 2\Omega \sin \varphi$, and Φ , Ω , a , λ , and φ are geopotential, rotational
angular velocity, earth radius, latitude, and longitude.
We have rechecked our wave flux and code. There is no problem with the wave
flux. Figure R6 shows the wave flux based on our code (upper) and the wave flux
in a previous study (bottom). These two figures show a similar wave flux pattern
with a similar magnitude. The strange arrows in our last version come from the
too small arrows and the scale of the graph. We have optimized these arrows (Fig.
R4e).

[REDACTED]

**Figure R6 Plumb wave flux averaged over Jan 19-24 2016. Upper:** From our code (JRA55).

**Bottom:** From Shi et al. 2016 (ERA-Interim)

Ref: Shi C., Jin X., Liu R. The differences in characteristics and applicability among three types of
Rossby wave activity flux in atmospheric dynamics[J]. Transactions of Atmospheric
Sciences, 2017, 40(6): 850-855.

Specific comments:

Lines 54-59: The authors should give a more specific review on the relationship
between the QBO and East Asian climate;

**Response: Thanks. For East Asia, most studies focus on the effects of QBO**
**in winter, rather than summer. We have added a review of the linkage between**

**QBO and East Asia and its mechanism (the last paragraph of Introduction).**

*‘Recent studies have found that weather and climate variations over East Asia in*
*winter are controlled by QBO to a certain extent³⁵⁻³⁶. For instance, the easterly*
*phase of QBO (EQBO) tends to bring a warm winter to East Asia, which is achieved*
*mainly by regulating the subtropical jet stream and the East Asian winter monsoon*
*(EAWM) ³⁵⁻³⁶. The QBO-induced temperature changes can modulate static stability*
*near the tropical tropopause and further the Madden-Julian Oscillation (MJO)³⁷. The*
*difference in precipitation anomalies between the EQBO and westerly phases of QBO*
*(WQBO) is approximately 70% in MJO phases 6–8 from southern China to Japan.*
*The linkage between QBO and precipitation in southeast China is controlled by the*
*Holton-Tan effect (QBO’s influence on the stratospheric polar vortex) in winter; a*
*significant QBO-precipitation relationship occurs mainly when the Holton-Tan effect*
*is weak³⁸. Compared to winter, the influences of QBO in summer are not receiving*
*enough research attention. Some studies have found that QBO signals exist in*
*precipitation in some regions of East Asia, such as the Huaihe River valley³⁹⁻⁴⁰ and*
*August precipitation in northern China⁴¹. However, we still do not know whether*
*QBO is important in the leading mode of EASP and its mechanism.’*

Lines 61-62: Why do you focus on the mechanisms after 1995?

**Response: There are two reasons. First, the dipole mode occurs after the**
**mid-1990s, whereas there is a tripole mode before about 1995. Second, the**
**QBO’s subtropical path is sensitive to the amplitude of QBO, and the amplitude**
**of QBO is significantly enhanced after the mid-1990s, which leads to more**
**obvious influences on the East Asian climate. Figure R7 shows the 21-year sliding**
**correlation coefficients between QBO and DMI (red) and the QBO amplitude**
**(orange). We can see there is a near-linear relationship between the correlation**
**coefficients and amplitude. The larger the QBO amplitude, the stronger the**
**QBO-precipitation relationship (Fig. R7b). Our next work will discuss this**
**process.**

 **Figure R7 Linkage between QBO amplitude and QBO-precipitation relationship. a** 21-year
 sliding correlation coefficients between QBO and DMI (red) and the QBO amplitude (orange). **b**
 Scatter diagram for QBO amplitude and QBO-DMI relationship.

Lines 85-88: The QBO index should be plotted in Figure 1d;

**Response. Thanks, we added the QBO index in Figure 1d (green line, Fig. R8).**

 **Figure R8, Figure 1d in the revised manuscript.**

Lines 110-111: These signs are too small to be distinguished;

**Response. Thanks, we enlarged these signs.**

Lines 121-122: The longitudes and latitudes of the NTA, SNA and SIO regions should
 be marked in Figures 2b-d;

**Response. Thanks. Dashed boxes show the location of the NTA, SNA, and SIO**
 **regions (Figure R9).**

Figure R9. Figure 2d-f in the revised manuscript.

Figure 3b: The wave flux vectors are too small;

**Response: Thanks, we enlarged these vectors.**

Lines 144-148: It is not Kelvin wave that drives the acceleration of the zonal winds.

The authors should present the EP flux and its divergence when talking about the
 acceleration of the zonal winds. The downward propagation of the QBO signal should
 be shown in a temporal evolution map.

**Response: Thanks. Here, we want to say that the Kelvin wave drives zonal wind
 anomalies (the QBO signal) in the tropical stratosphere, rather than the
 tropospheric westerly anomalies in the subtropics and the easterly anomalies in
 the mid-latitudes. The zonal wind anomalies over East Asia are forced by the
 Rossby wave. Here, we use Plumb wave flux to diagnose the propagation of the
 Rossby wave, because we show mainly a local wave-flow feedback process (Fig.
 R4b, e). The EP flux is a zonal mean result. We added the downward propagation
 of the QBO signal in the revised manuscript (Figure R10).**

Figure R10. Figure 4i in the revised manuscript.

Figure 5c: Please check the calculation of the wave flux;

**Response: We have rechecked the Plumb wave flux and its code. There is no**
**problem with the code. The strange arrows in our last version come from the too**
**small arrows and the scale of the graph. We have optimized these arrows (Fig.**
**R4e).**

Lines 151-153: Why did the temperature gradient anomalies can extend the QBO
wind signal to the troposphere?

**Response: We have removed this sentence because the refraction index is**
**difficult to diagnose in a local region. Temperature gradient anomalies can**
**influence the refraction index and further the propagation of the Rossby wave.**
**The wave-flow feedback extends and enhances the QBO signal in the**
**troposphere.**

Lines 153-156: What kind of wave-flow interaction can influence the wind anomalies
over the subtropics and mid-latitude? The authors should present more detailed
information on the generation of the negative wind anomalies over the mid-latitude.
The current statement is not convincing. Furthermore, please check the calculation of
the wave flux in Figure 3c;

**Response: It is the Rossby wave-flow feedback. The QBO-induced subtropical**
**wind anomalies disturb the Rossby wave source (Fig. R11c, e, color), leading to**
**the Rossby wave propagating northward (Fig. R11c, e, arrows). The Rossby wave**
**converges in the mid-latitudes and drives the easterly anomalies (Fig. R11b).**
**Note that such Rossby wave source and wave flux anomalies occur mainly in the**
**upper troposphere (above 300 hPa, Fig. R11c, e) and no such phenomenon is**
**observed at 500 hPa (Fig. R11g). This means that the wave-flow feedback is an**
**upper tropospheric process related to QBO. Figure 3c has been replaced by**
**Figure 3e in the revised manuscript (Figure R12), and we optimized the arrows.**
**We can see that the arrows depart from the positive Rossby wave source region**
**and propagate northward.**

Figure R11. (Supplementary Fig. 6) East Asian circulation anomalies regressed on the standardized 30-hPa QBO index. a Zonal wind (colors) and meridional circulation (vectors) anomalies averaged over 70°-120°E. **b** Plumb wave flux (vectors) and the $F\phi$ divergence (colors) anomalies averaged over 70°-120°E. **c** The 150-hPa Rossby wave source (colors) and Plumb wave flux (vectors) anomalies. **d** The 150-hPa zonal wind (colors) and geopotential height (contours) anomalies. **e** and **g** As in **c** but for 300 hPa and 500 hPa, respectively. **f** and **h** As in **d** but for 300 hPa and 500 hPa, respectively. The units of geopotential height are gpm in **d**, **f**, **h**. Linear effects of NTA, SNA, and SIO have been removed, and dots indicate values at the 95% confidence level.

**Figure R12.** (Fig. 3e in the revised manuscript) The 200-hPa Rossby wave source (colors) and
 Plumb wave flux (vectors) anomalies regressed on the standardized QBO index.

Lines 158-160: It is barotropic structure that extends from the upper troposphere to
 the middle troposphere, not downward extension;

**Response: It is ‘extend’. We have revised this sentence.**

Lines 160-162: Please rewrite this ambiguous sentence.

**Response: We removed this sentence and the QBO section has been rewritten.**

Reviewer #3 (Remarks to the Author):

This work analyzed a stratospheric precursor of East Asian summer droughts and
floods, which is the QBO. The authors indicated that the QBO could modulate the
East Asian precipitation via a subtropical way, contributing the largest explained
variation of the south-north dipole pattern of summer rainfall in East Asia. This is an
interesting and important topic, which provides a potential prediction signal for
seasonal prediction in China. Overall, the manuscript is well written, and the results
are credible. The contents appear suitable for publication, but some necessary revision
is needed.

1. In the “Methods” section, it is provided how to establish a multiple linear equation
associated with the dipole mode of summer rainfall, but each coefficient of the final
linear equation is not provided established through historical data. This should be one
of the most important results of this work, so it is suggested to be clear, which is also
an indicator that intuitively reflects the contribution of various variables. In addition,
the manuscript did not consider whether the selected indicators are independent of
each other, which should also be explained.

**Response: Thanks. We have added the coefficients in the revised manuscript**
**(Figure R1) and the multicollinearity. Table R1 shows the tolerance and the**
**variance inflation factor (VIF) among different factors. We can see that the VIF**
**is closer to 1.0 and less than 10, while the tolerance is closer to 1.0. Thus, the**
**multicollinearity is weak and the variables are approximately independent of**
**each other.**

**Table R1 (Supplementary Table 3) Multicollinearity of different factors**

	QBO _{10-hPa}	NTA	SNA	SIO	QBO _{70-hPa} res
Tolerance	0.903	0.716	0.897	0.912	0.769
Variance inflation factor	1.107	1.397	1.115	1.096	1.300

**Figure R1.** (Figure R5b in the revised manuscript), **b** DMI (colors) and its forecasting curve
 (black line).

2. The manuscript specifically evaluated the contributions of QBO in 2020 and 2021,
 and it is suggested that the specific values of the QBO index in 2020 and 2021 should
 be provided in the manuscript, or the time series of the QBO index is extended to
 2022 in Fig. S8.

**Response:** Thanks. We extended the QBO index to 2020 in Figure S8.

**Figure R2.** (Supplementary Fig. 8) **Downward extension of the QBO signal.** **a** Temporal
 development of the QBO index. **b** The QBO anomalies lag-regressed on the DMI index. **c** Time
 series of the 30-hPa QBO index in July-August and the 10-hPa QBO index in March. Straight
 lines in (b) are July-August, and dots indicate values at the 95% confidence level.

3. Readers may be curious that since QBO is quasi-biennial oscillation signal, and the
summer rainfall in 2020 and 2021 was significantly affected by QBO, why do the
July-August rainfall anomalies in 2020 and 2021 exhibited a consistent spatial
distribution of SDNF mode? The manuscript also mentioned that QBO has an impact
on the a typical north-south reverse mode in 2022, and how did this affect it? It is
suggested to add more discussions on these issues in the “Discussion” section.

**Response: Yes. It is a good suggestion. We have added some description in the**
**“Discussion” section. Although the period of QBO is two years on average, it is**
**not a simple constant. Figure R3 shows the QBO index from 2019 to 2022. The**
**easterly wind in 2020 continues a short time and a QBO disruption occurs in**
**2020-2021. Thus, the easterly wind appears for two consecutive years.**

**For 2022, we said that the forecasting model including QBO can rebuild the**
**droughts in some areas in 2022 (blue lines in Figure R4a, c), not that the QBO**
**plays a dominant role. The precipitation in 2022 is not a dipole mode near 30°N**
**(Figure R4a). Our statistical model is established according to the impacting**
**factors of the dipole mode over East Asia, and some factors influencing local**
**precipitation are not considered; thus, it can predict only some of the droughts**
**and floods in 2022. QBO does not play a dominant role in this case, but**
**contributes positively to the droughts between 27°N and 35°N (Figure R4b).**

**This work is theoretical research based on a single-level QBO index. Our**
**new work finds that QBO structure and amplitude are important (Fig. R5).**
**Figure R5b shows a scatter diagram for the QBO-precipitation relationship and**
**30-hPa QBO amplitude. We can see an evident linear linkage between QBO**
**amplitude and the QBO-precipitation relationship. The QBO amplitude is**
**depend on the whether the peak in a QBO period is near the 30 hPa. We will**
**display these results in a future work. In addition, East Asia is a large region, and**
**the major impacting factors may vary by subregion. We may need a more**
**detailed forecasting model.**

Figure R3. Development of the QBO index.

Figure R4. (Supplementary Fig. 10d-f) a Precipitation anomalies in 2022. b QBO-induced precipitation anomalies in 2022. c Forecasting precipitation anomalies in 2022.

Figure R5. Linkage between 30-hPa QBO amplitude and QBO-precipitation relationship. a 21-year sliding correlation coefficients between QBO and DMI (red) and the QBO amplitude (orange). b Scatter diagram for QBO amplitude and QBO-DMI relationship.

4. In the “Data” section, the materials and climate models used should have
corresponding references. For example, the West Pacific subtropical high indices can
be supplemented with the following references:

**Response: Thanks for your information. We have cited these papers in the**
**revised manuscript.**

Reconstruction and application of the monthly western Pacific subtropical high indices (in
Chinese). J. Appl. Meteor. Sci., 2021, 23: 414-423.

The Asian Summer Monsoon: Characteristics, Variability, Teleconnections and Projection. 2019,
pp 1-237. doi: 10.1016/C2017-0-04074-0. Elsevier.

Danabasoglu, Gokhan, et al. "The community earth system model version 2 (CESM2)." Journal of
Advances in Modeling Earth Systems 12.2 (2020): e2019MS001916.

REVIEWER COMMENTS

Reviewer #1 (Remarks to the Author):

The authors have well addressed the concerns raised by the referees. I think this work can be accepted for publication in the present form.

Reviewer #2 (Remarks to the Author):

My previous questions have been addressed by the authors, which is much appreciated. However, I still have some concerns about the authors explanation of the physical process, one of them is the wave-flow interaction interpretations in Figure 3, which is vital to verify the topic of this manuscript. The authors should check or give more physical diagnose evidence to support their statement.

Lines 106-107: It is better to mark the location of the WNPSH with contours;

Lines 182-184: More detailed information should be given on the process that the westerly anomalies induce positive RWS anomalies.

Lines 184-187: The divergence or convergence of the Rossby waves is not directly linked to the changes in the zonal winds. I can not agree with the statement in this sentence.

**REVIEWER COMMENTS**

Reviewer #2 (Remarks to the Author):

My previous questions have been addressed by the authors, which is much
appreciated. However, I still have some concerns about the authors explanation of the
physical process, one of them is the wave-flow interaction interpretations in Figure 3,
which is vital to verify the topic of this manuscript. The authors should check or give
more physical diagnose evidence to support their statement.

Lines 106-107: It is better to mark the location of the WNPSH with contours;

**Response:** Thank you. We added the contours of WNPSH. The Fig. 2a is reorganized
as follows:

**Figure R1 (Figure 2a in the revised manuscript).** Black and red contours in a are
the 5880-gpm geopotential height (WNPSH) during SFND and SDF years,
respectively.

Lines 182-184: More detailed information should be given on the process that the
westerly anomalies induce positive RWS anomalies.

**Response:** Thank you. The westerly anomalies in the subtropical region mean a

meridional gradient of zonal wind ($\frac{\partial u}{\partial y}$) anomalies there, which further influence the

relative vorticity ($\zeta = \frac{\partial v}{\partial x} - \frac{\partial u}{\partial y}$). The meridional advection of absolute vorticity
 ($\zeta + f$) and meridional vortex stretching contribute the positive RWS anomaly, based
 on the RWS equation ($RWS = -\nabla \cdot [V_{\chi}(\zeta + f)] = -\frac{\partial[u_{\chi}(\zeta + f)]}{\partial x} - \frac{\partial[v_{\chi}(\zeta + f)]}{\partial y}$).

Such process is rephrased as follows: ‘According to the equation of relative vorticity
 ($\zeta = \frac{\partial v}{\partial x} - \frac{\partial u}{\partial y}$), meridional gradient of zonal wind will increase (reduce) the relative
 vorticity on the north (south) side of these subtropical westerly anomalies (Fig. 3e).
 The abnormal relative vorticity could disturb the anomalous meridional vortex
 stretching and meridional advection of absolute vorticity term of RWS (see Fig. 3f and
 method), which further cause a positive RWS anomaly in the subtropical region and
 poleward Rossby wave flux anomalies (Fig. 3g).’

 **Figure R2 (Figure 3e-h in the revised manuscript).** e The 200-hPa relative vorticity
 anomalies regressed on the standardized QBO index. f The meridional term of
 200-hPa Rossby wave source. g The 200-hPa Rossby wave source (colors) and Plumb
 wave flux (vectors) anomalies regressed on the standardized QBO index. h The
 200-hPa zonal wind (colors) and geopotential height (contours) anomalies regressed

on the standardized QBO index. The units of geopotential height are gpm in **h**. Linear
effects from NTA, SNA, and SIO have been removed in the anomalies in **b-h**. Dots
indicate values at the 95% confidence level.

Lines 184-187: The divergence or convergence of the Rossby waves is not directly
linked to the changes in the zonal winds. I cannot agree with the statement in this
sentence.

**Response:** Thank you for pointing this out. According to the kinematic equation,
zonal wind is directly related to the momentum flux divergence. We have rephrased
this sentence: *'The wave-induced momentum fluxes anomalously diverge in the*
*subtropics and converge in the mid-latitudes (Supplementary Fig. 6b), amplifying the*
*subtropical westerly anomalies and forcing mid-latitude easterly anomalies (colors,*
*Fig. 3h).'*

REVIEWERS' COMMENTS

Reviewer #2 (Remarks to the Author):

The authors have addressed my concerns, I do not have further comments.